

# Effect of extracts from eggs of *Helix aspersa maxima* and *Helix aspersa aspersa* snails on Caco-2 colon cancer cells

Magdalena Matusiewicz[1], Karolina Marczak[1], Barbara Kwiecińska[1], Julia Kupis[1], Klara Zglińska[2], Tomasz Niemiec[2] and Iwona Kosieradzka[2]

[1] Department of Nanobiotechnology, Institute of Biology, Warsaw University of Life Sciences, Warsaw, Poland
[2] Department of Animal Nutrition, Institute of Animal Sciences, Warsaw University of Life Sciences, Warsaw, Poland

Corresponding author
Magdalena Matusiewicz,
magdalena_matusiewicz@sggw.edu.pl

## ABSTRACT

**Background:** Colorectal cancer is the third most commonly diagnosed cancer. Natural compounds, administered together with conventional chemotherapeutic agent(s) and/or radiotherapy, may be a novel element in the combination therapy of this cancer. Considering the anticancer properties of compounds derived from different tissues of various snail species confirmed earlier, the purpose of the present research was to evaluate the effect of extracts from eggs of *Helix aspera maxima* and *Helix aspersa aspersa* snails, and fractions of extracts containing particles of different molecular weights on Caco-2 human epithelial colorectal adenocarcinoma cells.
**Methods:** The extracts and fractions were analyzed for antioxidant activity, phenols and total carbohydrates using colorimetric methods. Lipid peroxidation products and glutathione in eggs were also examined using these methods. Crude protein and fat in eggs were determined. Molecular weights of egg proteins and glycoproteins were analyzed by sodium dodecyl sulfate-polyacrylamide gel electrophoresis. Astaxanthin, selected vitamins and amino acids in eggs were measured using liquid chromatography methods, and minerals by emission spectroscopy, mass spectrometry or X-ray fluorescence. The action of extracts on the cell viability was determined by the MTT (methylthiazolyldiphenyl-tetrazolium bromide) test, based on the mitochondrial oxidative activity, after 24 and 72 h of treatment. The influence of fractions on the cell viability was assayed after 24 h. The effect of extracts on the percentage of live and dead cells was evaluated by the trypan blue assay, in which live cells exclude trypan blue, while dead cells take up this dye, after 12, 24, 48 and 72 h of treatment. Their influence on the integrity of cell membranes was determined based on the activity of LDH (lactate dehydrogenase), released from damaged cells, after 24 and 72 h of treatment. Then, the effect of extracts on the content of lipid peroxidation products in cells was examined using colorimetric method, after 24 h of treatment. Their influence on types of cell death was determined by flow cytometry, after this time.
**Results:** The extracts and their fractions containing molecules <3 kDa decreased the cell viability, after 24 h of treatment. The extracts reduced the percentage of live cells (also after 48 h), increased the degree of cell membrane damage and the amount of lipid peroxidation products, induced apoptosis and reduced necrosis.

**How to cite this article** Matusiewicz M, Marczak K, Kwiecińska B, Kupis J, Zglińska K, Niemiec T, Kosieradzka I. 2022. Effect of extracts from eggs of *Helix aspersa maxima* and *Helix aspersa aspersa* snails on Caco-2 colon cancer cells. **PeerJ** 10:e13217 **DOI** 10.7717/peerj.13217

**Conclusions:** Antioxidants, phenols, lipid peroxidation products, anticancer peptides, restriction of methionine, appropriate ratio of essential amino acids to non-essential amino acids, vitamin $D_3$, Ca, Mg, S, Cu, Mn, Zn, Se and other bioactive compounds comprised in the extracts and their additive and synergistic effects may have influenced Caco-2 cells. Natural extracts or the chemical compounds contained in them might be used in the combination therapy of colorectal cancer, which requires further research.

## INTRODUCTION

Colorectal cancer is the third most commonly diagnosed cancer and the second leading cause of cancer death (*Bray et al., 2018*). The incidence rates of colorectal cancer are over three-fold greater in countries that have completed their socioeconomic transition, with high/very high Human Development Index (HDI) comparing to transitioning countries, with low/medium HDI while the mortality rates are over two-fold greater (*Bray et al., 2018*). The greatest colon cancer incidence rates were found in Europe, Australia/New Zealand, Northern America, Uruguay and Eastern Asia (*Bray et al., 2018*).

Colorectal cancer occurrence and progression are caused by many risk factors, the main of which are age, gender, family and personal history, and region (*Huang et al., 2019b*). The risk of this cancer can be reduced through proper dietary habits and lifestyle (*Rejhová et al., 2018*).

The main treatment for patients suffering from a potentially curable colorectal cancer is surgery, and chemotherapy and/or radiation therapy may be given before/after it, depending on the stage of the disease (*Redondo-Blanco et al., 2017*). However, this treatment is not sufficient to control colorectal cancer as 30% of stage I to III patients and up to 65% of stage IV patients relapse, underlining the urgent need to find more effective therapies. Adverse effects of the above treatment impair the life quality, may unfavorably influence on the course, outcomes and costs of treatment (*Rejhová et al., 2018*).

Treatment outcomes and life quality can be improved by application of natural products derived from plants, animals and microorganisms, well tolerated and less toxic than conventional chemotherapeutic agents (*Huang et al., 2019b*). There are nine anticancer drugs from marine organisms on the market and other molecules from these organisms are being examined as anticancer drugs in different phases of clinical trials (*Dyshlovoy & Honecker, 2020*).

Natural compounds, administered together with conventional chemotherapeutic agent(s) and/or radiotherapy, may be a novel element in combination therapy of colorectal cancer (*Rejhová et al., 2018*; *Redondo-Blanco et al., 2017*). Certain compounds may sensitize to cytotoxic therapy, intensify the effective concentration of a drug, enhance the combined effects of both therapeutics or exert a cytotoxic effect specifically on cancer cells.

Moreover, combination therapy targets many signaling pathways and uses a variety of mechanisms to decrease the development of anticancer drug resistance.

Snails can provide many bioactive compounds for the pharmaceutical and cosmetics industries, applicable in the development of novel preparations with lower toxicity and subsequent effects in comparison with compounds commonly used for this purpose (*Dhiman & Pant, 2020*). Information on the chemical composition and nutritional value of mucus, foot tissues and shells of one of the most popular edible land snails of the subspecies *Helix aspersa aspersa* is presented in the article by *Matusiewicz et al. (2018)*. The use of water extracts from lyophilized mucus and foot tissues of these snails reduced the viability of Caco-2 human colorectal adenocarcinoma cells.

*In vitro* experiment of *Ellijimi et al. (2018)* demonstrated that the mucus extract derived from *Helix aspersa maxima* snails decreased the content of melanin and tyrosinase activity in B16F10 murine melanoma cells and IGR-39 and SK-MEL-28 human melanoma cells. It decreased the viability of human melanoma cells and did not influence on the viability of HaCaT human keratinocytes. It induced a caspase-dependent apoptosis of human melanoma cells, inhibited their migration and invasion by decreasing the production of matrix metalloproteinase-2. It strongly affected the adhesion of IGR-39 cells by blocking $\alpha_2\beta_1$ and $\alpha_v\beta_3$ integrin functions and by decreasing the production of $\alpha_v$ and $\beta_1$ integrins. The mucus and its two fractions derived from *Achatina fulica* snail reduced the viability of MCF-7 breast cancer cells and Vero epithelial cells (*Teerasak et al., 2016*).

Water extract from *H. a. aspersa* snails exhibited anticancer activity against Hs578T breast cancer cells (*El Ouar et al., 2017*). It induced necrosis of cancer cells, stimulated the mRNA expression of tumor necrosis factor (TNF)-$\alpha$ and inhibited the expression of nuclear factor kappa-light-chain-enhancer of activated B cells (NF-κB), phosphatase and tensin homolog (PTEN) and tumor protein 53 (p53).

Many studies showed that mollusk and arthropod hemocyanins, oxygen-transporting hemolymph glycoproteins, have significant anticancer activity which was demonstrated in both *in vitro* and *in vivo* models. Hemocyanins have immunostimulatory activity, thanks to the structural properties, they stimulate the immune system nonspecifically, they interact with macrophages, granulocytes, CD4+ and CD8+ cells, induce potent cellular and humoral immune responses (*Antonova et al., 2014*; *Dolashka et al., 2015*; *Mora et al., 2019*; *Dolashka et al., 2011*; *Salazar et al., 2019*; *Dolashki et al., 2019*; *Dolashka-Angelova et al., 2008*; *Stenzl et al., 2016*; *Antonova et al., 2015*; *Guncheva et al., 2020a*, *2020b*; *Georgieva et al., 2020*). The structural subunits of hemocyanins derived from the land snails *H. a. aspersa* and *Helix lucorum*, the marine snail *Rapana venosa* and the mucus of *H. a. aspersa* snails reduced the viability of HT-29 human colorectal adenocarcinoma cells (*Georgieva et al., 2020*). The mucus and the α and βc subunits of *H. a. aspersa* hemocyanin decreased the viability to the greatest extent. The half-maximal inhibitory concentrations (IC$_{50}$) of above preparations for HT-29 cells were lower than for Balb/c3T3 fibroblasts. The mechanisms of their anticancer activity included apoptosis. The hemocyanins isolated from the land snail *Helix pomatia* and the marine snail *Rapana thomasiana* demonstrated strong anticancer and antiproliferative actions in a colon

carcinoma murine model (*Gesheva et al., 2014*). The immunization with these hemocyanins resulted in prolonged survival of animals, improved humoral anticancer response, inhibition of tumor growth, splenomegaly and appearance of lung metastasis. Treatment with hemocyanins derived from *H. pomatia* and *R. thomasiana* in other study resulted in the production of large amounts of antitumor IgGs, plasma cells as well as tumor specific cytotoxic T cells, stimulation of the secretion of proinflammatory cytokines, suppression of tumor size and growth, and prolongation of the life span of a colon carcinoma murine model (*Stoyanova et al., 2020*).

Considering the above anticancer properties of compounds derived from different tissues of various snail species, the extracts from snail eggs, containing chemical compounds characterized by potential anticancer activities, may affect the growth and development of cancer cells. The purpose of the research was to evaluate the chemical composition of eggs from the popular edible farm snails *Helix aspera maxima* and *Helix aspersa aspersa*, water extracts from these eggs and fractions of extracts containing particles of different molecular weights. The action of extracts and their fractions on the viability of Caco-2 human epithelial colorectal adenocarcinoma cell line was determined. The effect of the extracts on the percentage of live and dead cells, the integrity of cell membranes, the content of lipid peroxidation products in cells and the types of cell death was then examined.

## MATERIALS AND METHODS

### Animal material and preparation of lyophilizates

Two-day-old eggs (about 0.5 kg) of two snail subspecies, *Helix aspersa maxima* Taylor, 1883 (*Cornu aspersum maxima* (Taylor, 1883)) and *Helix aspersa aspersa* Müller, 1774 (*Cornu aspersum aspersum* (Müller, 1774)), were obtained from the commercial breeding (Grudziądz, Poland). Animal raw materials were collected in June 2018.

Fresh eggs, a few hours after harvest, were washed, homogenized and frozen at −80 °C, for 2 days. Then, the raw materials were lyophilized (Lyovac GT 2 freeze-dryer; SRK Systemtechnik GmbH, Riedstadt, Germany) for 2 days, in the dark. Lyophilized eggs were milled into fine powder using a laboratory mill and stored in polypropylene tubes (−80 °C).

### Preparation of extracts and their fractions for the determination of antioxidant indicators, phenols and total carbohydrates

Before each analysis, lyophilized eggs derived from *H. a. maxima* and *H. a. aspersa* snails were homogenized in deionized water (concentration 100 mg/mL), by vortexing. The samples were left for extraction (30 min, 4 °C), subjected to centrifugation (1,600× *g*, 10 min) and the supernatants (extracts) were collected. Then, the extracts were filtered using polyvinylidene fluoride (PVDF) syringe filters (pore size 0.22 μm; EuroClone, Pero, Italy). The extracts were fractionated, based on the molecular weight, using ultra centrifugal filter devices containing regenerated cellulose membranes (Merck Millipore, Burlington, MA, USA), including 3, 10 and 50 kDa cutoffs, in compliance with the manufacturer's prescriptions regarding g-force and centrifugation time. Four fractions

were obtained: >50 kDa (>50 K), 10–50 kDa (10–50 K), 3–10 kDa (3–10 K) and <3 kDa (<3 K).

## Determination of antioxidant indicators
### Ferric-reducing antioxidant power

Ferric-reducing antioxidant power of extracts and four fractions: >50 K, 10–50 K, 3–10 K and <3 K of extracts from lyophilized eggs of *H. a. maxima* and *H. a. aspersa* snails was determined using the modified Oyaizu method (*Oyaizu, 1986*; *Matusiewicz et al., 2019*). It involves reduction of $Fe^{3+}$, being in stoichiometric excess over antioxidants, because of electron donation by them. The absorbance increase is noted as the reduction capability is higher. Properly diluted extracts and fractions (2.5 mL) were mixed with 0.2 M sodium phosphate buffer pH 6.6 (2.5 mL) and 1% potassium ferricyanide (2.5 mL). The samples were incubated at 50 °C for 20 min and then 10% trichloroacetic acid (TCA, 2.5 mL) was added. The probes were centrifuged ($3,000\times g$, 5 min) and the supernatants (0.4 mL) were combined with deionized water (0.4 mL) and 0.1% ferric chloride (160 µL). The absorbance was measured at 700 nm, using microplate reader (Infinite M200; Tecan, Männedorf, Switzerland). The standard curve was generated by applying various concentrations, ranging 0–100 µM, of (±)-6-hydroxy-2,5,7,8-tetramethylchromane-2-carboxylic acid (TROLOX), a water-soluble analog of vitamin E. The assay was conducted in three replicates ($n = 3$).

### 2.2′-azino-bis(3-ethylbenzthiazoline-6-sulfonic acid) radical cation (ABTS·+) scavenging activity

The relatively stable ABTS·+ discolors when is reduced. To determine ABTS·+ scavenging activity of extracts and four fractions of extracts from *H. a. maxima* and *H. a. aspersa* eggs, the procedure of *Sun et al. (2007)* with modifications was used (*Matusiewicz et al., 2019*). This method is based on the addition of antioxidants to ABTS·+ solution and spectrophotometric determination of the remaining ABTS·+. To make ABTS reagent, 7 mM ABTS (5 mL) was combined with 140 mM $K_2S_2O_8$ (88 µL). To generate free radicals, the mixture was put in the dark (16 h, room temperature (RT)). The reagent was diluted using the absolute ethanol, to get the absorbance of $0.70 \pm 0.02$ at 734 nm (Infinite M200 microplate reader; Tecan, Männedorf, Switzerland). ABTS·+ scavenging activity was assayed by combining of ABTS reagent (0.9 mL) with properly diluted extracts or fractions of extracts from snail eggs (0.1 mL). After the incubation of samples (6 min, RT), the absorbance was registered. The standard curve was obtained using TROLOX (see Ferric-reducing antioxidant power section), $n = 3$.

### 2.2-diphenyl-1-picrylhydrazyl radical (DPPH·) scavenging activity

To evaluate DPPH· scavenging activity of extracts and fractions of extracts from *H. a. maxima* and *H. a. aspersa* eggs, the method of *Li, Zhang & Wang (2012)* with some modifications was utilized (*Matusiewicz et al., 2019*). The procedure is based on donation of hydrogen atom or electron by antioxidants to the unpaired electron of DPPH·, the absorbance falls proportionally to rise of DPPH non-radical form. 0.2 mM DPPH· in absolute methanol was mixed with properly diluted extracts and fractions of extracts from

eggs (2:1, v/v). After incubation of the samples (30 min, without access to light), they were centrifuged (15,000× $g$, 10 min). Thereafter, the supernatant absorbance was registered at 517 nm (Infinite M200 microplate reader; Tecan, Männedorf, Switzerland). The standard curve was obtained as in Ferric-reducing antioxidant power section, $n = 3$.

## Determination of phenols

The method of determination of total phenols, the Folin-Ciocalteu method, is based on the reading of the absorbance of the complex arising from the reduction of Folin-Ciocalateu reagent, *i.e.*, salts of hetero polyacids, phosphomolybdic and phosphotungstic (*Matusiewicz et al., 2019*; *Singleton, Orthofer & Lamuela-Raventós, 1999*). Over the reaction Mo (VI) ions are reduced to Mo (V) ions, resulting in a blue color of $[PMoW_{11}O_{40}]^{4-}$. To evaluate the concentration of phenols, properly diluted extracts and fractions of extracts derived from snail eggs (0.5 mL) were mixed with Folin-Ciocalteu reagent (diluted 1:10 in deionized water, 2.5 mL). After incubation for 2 min, the samples were mixed with 7.5% $Na_2CO_3$ (2 mL) and incubated in a water bath (50 °C, 10 min). Then, the absorbance was read at 760 nm (Spectronic 20D cuvette spectrophotometer; Milton Roy, Rochester, NY, USA). The standard curve was obatained using various quercetin levels (0–100 µg/mL), $n = 2$.

## Determination of total carbohydrates, crude protein and crude fat

Concentration of total carbohydrates in extracts and fractions of extracts from *H. a. maxima* and *H. a. aspersa* eggs was measured using the phenol-sulfuric acid method (*Matusiewicz et al., 2019*; *Dubois et al., 1956*). The absorbance was registered at 490 nm and glucose was utilized as the standard, $n = 3$.

Content of crude protein in lyophilized eggs of *H. a. maxima* and *H. a. aspersa* snails was determined by the Kjeldahl method, according to AOAC International (*Matusiewicz et al., 2018*; *AOAC International, 2012*), $n = 2$.

Crude fat was extracted from lyophilized snail eggs by applying petroleum ether using the Soxhlet method (*Matusiewicz et al., 2018*; *AOAC International, 2012*), $n = 2$.

## Determination of lipid peroxidation products

Level of lipid peroxidation products – thiobarbituric acid reactive substances (TBARS) in extracts from lyophilized eggs of *H. a. maxima* and *H. a. aspersa* snails was determined using the procedure of *Uchiyama & Mihara (1978)* (*Matusiewicz et al., 2018*). Extracts were acquired after homogenization of egg lyophilizates in radioimmunoprecipitation assay (RIPA) buffer and centrifugation (1,600× $g$, 10 min). The absorbance was recorded at 532 nm (Infinite M200 microplate reader; Tecan, Männedorf, Switzerland). TBARS were expressed as the equivalents of malondialdehyde (MDA) and its precursor - 1.2.3.3-tetraethoxypropane (TEP) was used as the standard, $n = 6$.

## Determination of glutathione

Widespread in cells thiol tripeptide, glutathione (GSH), constitutes nearly 97% of non-protein thiol compounds. GSH is evaluated quantitatively by assay of non-protein -SH groups in the samples deproteinized by TCA. The method for the evaluation of

non-protein -SH groups consists in the Ellman's method, reduction of 5,5′-dithiobis (2-nitrobenzoic acid) (DTNB) by thiol compounds and formation of colorful 2-nitro-5-mercaptobenzoic acid, having an absorbance maximum at 412 nm (*Matusiewicz et al., 2019*; *Ellman, 1959*). Lyophilized eggs of *H. a. maxima* and *H. a. aspersa* snails were subjected to homogenization in 0.1 M phosphate buffer pH 7.4 and centrifugation (1,600× *g*, 10 min). In order to deproteinize, to the supernatants (1.5 mL) was added 50% TCA (78.96 µL) and samples were subjected to centrifugation (3,000 rpm, 5 min). Then, deproteinized supernatants (25 µL) were mixed with 0.2 M phosphate buffer pH 8.0 (200 µL) and with $6 \times 10^{-3}$ M DTNB (25 µL), directly on a 96-well plate. The absorbance was read by applying microplate reader (Infinite M200; Tecan, Männedorf, Switzerland). The standard curve was generated using different contents (0–75 nmol/mL) of GSH in 2.5% TCA, *n* = 6.

## Determination of astaxanthin and vitamins A, D$_3$, E and C

Determination of the contents of astaxanthin and vitamins A, D$_3$ (cholecalciferol) and E in lyophilized eggs of *H. a. maxima* and *H. a. aspersa* snails was preceded by liquid extraction with the application of ultrasound. The concentrations of above compounds were measured using high-performance liquid chromatography coupled with UV-Vis detection (HPLC/UV-Vis), based on the standard curves. The analysis of astaxanthin was performed using the Altus A-10 system (PerkinElmer, Waltham, MA, USA) and LiChroCART 250-4, C18 column (Merck & Co., Inc., Kenilworth, NJ, USA). The analyses of vitamins A and E were carried out by Shimadzu HPLC/UV-Vis system (Kyoto, Japan) and LiChroCART 125-4, C18 column (Merck & Co., Inc., Kenilworth, NJ, USA). The analysis of vitamin D$_3$ was done using the Altus A-10 system (PerkinElmer, Waltham, MA, USA) and LiChroCART 125-4, C18 column (Merck & Co., Inc., Kenilworth, NJ, USA). The analyses were carried out in a private analytical laboratory (Olsztyn, Poland), *n* = 3.

The analysis of vitamin C, as the sum of ascorbic acid and dehydroascorbic acid, in lyophilized snail eggs was carried out using HPLC/UV-V is, in the laboratory which is accredited by the Polish Centre for Accreditation (*PB 13 wydanie 6 z dnia 06.03.2012 r, 2012*). In order to determine the total content of vitamin C in the sample extracts, the reduction of dehydroascorbic acid to ascorbic acid was performed with dithiothreitol. The analysis was carried using a chromatograph with Waters® 2487 Dual Wavelength Absorbance Detector (Waters Corp., Miliford, MA, USA) with a Symmetry C18 column, 100 Å, 5 µm, 4.6 mm × 150 mm (Waters Corp., Miliford, MA, USA), column temperature: 25 °C, injection volume: 25–30 µl. The analysis was carried out at 245 nm wavelength and 0.8 mL/min mobile phase flow, *n* = 2.

## Analysis of molecular weights of proteins and glycoproteins

For the preparation of protein extracts, lyophilized eggs of *H. a. maxima* and *H. a. aspersa* snails were homogenized in phosphate-buffered saline (PBS) with inhibitors of proteases and phosphatases (Sigma-Aldrich, St. Louis, MO, USA), in the ratio 10 mg lyophilizate/

1 mL PBS – for analysis of proteins or 119 mg lyophilizate/1 mL PBS – for analysis of glycoproteins, and then centrifuged (1,600× $g$, 10 min). For protein profile analysis, the total protein concentration was equalized between extracts. Samples were submitted for sodium dodecyl sulfate-polyacrylamide gel electrophoresis (SDS-PAGE), with the application of a 5% stacking gel and 12% resolving gel – for separation of proteins or 10% resolving gel – for separation of glycoproteins, by the Laemmli method (*Matusiewicz et al., 2018*; *Laemmli, 1970*) with modifications. The samples (30 μL) were subjected to denaturation and reduction using the Laemmli sample buffer (Bio-Rad Laboratories, Hercules, CA, USA) with β-mercaptoethanol (30 μL) and then heating (95 °C, 5 min). Each sample (20 μL) and a protein marker (5 μL, PageRuler™ Plus Prestained Protein Ladder, 10 to 250 kDa; Thermo Fisher Scientific, Waltham, MA, USA – for analysis of proteins or ColorBurst™ Electrophoresis Marker; Sigma-Aldrich, St. Louis, MO, USA – for analysis of glycoproteins) were loaded onto two gels and resolved by electrophoresis (Mini-PROTEAN® Tetra Vertical Electrophoresis Cell System; Bio-Rad Laboratories, Hercules, CA, USA). In the case of glycoprotein analysis, a postive control, 10 μL of horseradish peroxidase at a concentration of 2 mg/mL with the Laemmli sample buffer, was also used.

To analyze the molecular weights of proteins, protein bands separated on the first gel were fixed, stained (QC Colloidal Coomassie Stain; Bio-Rad Laboratories, Hercules, CA, USA) and destained in accordance with the manufacturer's procedure (Bio-Rad Laboratories, Hercules, CA, USA).

Sugar moieties of glycoproteins separated on the second gel were detected by applying the commercial kit, according to the manufacturer's procedure (Pierce Glycoprotein Staining Kit; Thermo Fisher Scientific, Waltham, MA, USA). Glycols present in glycoproteins are oxidized to aldehydes when treated with periodic acid and are stained magenta.

The gels were visualized by applying Azure c400 imaging system (Azure Biosystems, CA, USA).

## Analysis of amino acids

The concentrations of the amino acids with the exception of tryptophan (Trp) in lyophilized eggs of *H. a. maxima* and *H. a. aspersa* snails were assayed by ion-exchange chromatography with spectrophotometric detection (IEC-VIS) (*Annex III F, 2009*). The analysis was performed using an automatic amino acid analyzer AAA 400 and an ion-exchange column (Ingos, Prague, Czech Republic).

The level of Trp was determined using high-performance liquid chromatography with fluorescence detection (HPLC-FLD, Agilent 1100 Series; Agilent Technologies, Santa Clara, CA, USA) (*Matusiewicz et al., 2018*; *Annex III G, 2009*). Zorbax® ODS C18, 4.6 mm ID × 250 mm (5 μm) column (Agilent Technologies, Santa Clara, CA, USA) was used.

Analysis of amino acids was carried out in the laboratory accredited by the Polish Centre for Accreditation, $n = 2$.

The amino acid score (AAS), chemical score (CS) and essential amino acid index (EAAI) were calculated using the equations below (*Matusiewicz et al., 2018*; *FAO/WHO, 1991*; *Oser, 1959*):

$$AAS = \frac{aa}{AA(standards)} \tag{1}$$

$$CS = \frac{aa}{AA(egg)} \tag{2}$$

$$EAAI = \sqrt[n]{\frac{100A}{AS} \times \frac{100B}{BS} \times \frac{100C}{CS} \times ... \times \frac{100H}{HS}} \tag{3}$$

in which aa – the content of amino acid in the protein of snail eggs (%); AA (standards) – the content of amino acid in the reference protein for 2–5 years old children (%) (*FAO/WHO, 1991*); AA (egg) – the content of amino acid in the whole egg reference protein (%) (*Oser, 1959*); n – amino acid number; A, B, C, H – the content of EAA (essential amino acids) in the protein of snail eggs (%); AS, BS, CS, HS – the content of EAA in the reference protein (%).

## Analysis of minerals

The contents of Ca, P, Na, K, Mg, Cu, Fe, Mn and Zn in lyophilized eggs of *H. a. maxima* and *H. a. aspersa* snails were assayed by inductively coupled plasma—atomic emission spectroscopy (ICP-AES, iCAP 6500; Thermo Fisher Scientific, Waltham, MA, USA) (*PB 34 wydanie 7 z dnia 08.03.2017 r, 2017*). The concentrations of Ni, Cr, Mo, B, Co, Se, V and Sn in snail eggs were determined by mass spectrometry with ionization in inductively coupled plasma (ICP-MS, Varian 820-MS; Varian, Inc., Palo Alto, CA, USA) (*CLA/ESA/5/2014 wersja 2 z dnia 03.03.2014 r, 2014*). The contents of S, Cl, Si, I and F in eggs were evaluated by wavelength-dispersive X-ray fluorescence (WDXRF, Axios, PANalytical, Almelo, the Netherlands) (*CLA/ESA/3/2014 wersja 1 z dnia 03.03.2014 r, 2014*). The analyses were carried out in the laboratories accredited by the Polish Centre for Accreditation, *n* = 3.

## Preparation of extracts and their fractions for cell culture tests

Before cell culture tests, lyophilized eggs derived from *H. a. maxima* and *H. a. aspersa* snails were homogenized in deionized water, at the concentration of 25 mg/mL ((tests described in Effect of extracts on cell viability (MTT test), Effect of extracts on the percentage of live and dead cells (trypan blue test), Effect of extracts on the integrity of cell membranes and Effect of extracts on the types of cell death sections)) or 2.5 mg/mL (Effect of extracts on the content of lipid peroxidation products section). The homogenates were left for extraction (30 min, 4 °C), centrifuged (1,600× *g*, 10 min) and the extracts were collected.

For the experiment described in Effect of fractions of extracts on cell viability (MTT test) section, the extracts at the concentration of 100 mg/mL were prepared, filtered and fractionated to obtain four fractions: >50 K, 10–50 K, 3–10 K and <3 K, as in Preparation of

extracts and their fractions for the determination of antioxidant indicators, phenols and total carbohydrates section.

Extracts or fractions were sterilized (PVDF syringe filters, pore size 0.22 μm; EuroClone, Pero, Italy) under the biological safety cabinet (TopSafe™ 1.2, class II; BIOAIR, Pavia, Italy). Then, for some tests, appropriate dilutions were prepared with sterile deionized water.

## Caco-2 cell culture

Human epithelial colorectal adenocarcinoma (Caco-2) cell line (ECCC, 55 passage; Sigma-Aldrich, St. Louis, MO, USA) was cultivated in polystyrene plates designed for adherent cell culture ((for the experiments described in Effect of extracts on cell viability (MTT test), Effect of fractions of extracts on cell viability (MTT test) and Effect of extracts on the integrity of cell membranes sections – in 96-well plates, at an initial density of $1 \times 10^4$ cells/100 μL, for the experiment presented in Effect of extracts on the percentage of live and dead cells (trypan blue test) section – in 24-well plates, at a density of $5.94 \times 10^4$ cells/594 μL and for the experiments described in Effect of extracts on the content of lipid peroxidation products and Effect of extracts on the types of cell death sections – in 6-well plates, at a density of $0.75 \times 10^5$ cells/1.5 mL)) in Minimum Essential Medium (MEM) comprising 2 mM L-glutamine, 10% fetal bovine serum (FBS), 1% non-essential amino acids (NEAA) and 1% antibiotic-antimycotic (all solutions were purchased from Thermo Fisher Scientific, Waltham, MA, USA) (*Matusiewicz et al., 2018*; *Matusiewicz et al., 2019*). The cells were kept at 37 °C in 5% $CO_2$ and 95% relative humidity in a $CO_2$ incubator (INCO 108 med; Memmert GmbH + Co. KG, Schwabach, Germany) for 24 h. After incubation and reaching about 70% confluency, they were starved in MEM with 1% FBS and 1% antibiotic-antimycotic (Thermo Fisher Scientific, Waltham, MA, USA) overnight (*Matusiewicz et al., 2018*; *Matusiewicz et al., 2019*).

## Effect of extracts on cell viability (MTT test)

A total of 90 μL of new medium (MEM with 1% FBS and 1% antibiotic-antimycotic; Thermo Fisher Scientific, Waltham, MA, USA) and 10 μL of extracts from lyophilized eggs of *H. a. maxima* and *H. a. aspersa* snails, at the concentrations of 25, $25 \times 10^{-1}$, $25 \times 10^{-2}$, $25 \times 10^{-3}$, $25 \times 10^{-4}$ and $25 \times 10^{-5}$ mg/mL were added to the cells. Equal volumes of deionized water (sterile) were introduced into the control cells. Additional controls were also included. After 24 h and 72 h of incubation in a $CO_2$ incubator (INCO 108 med; Memmert GmbH + Co. KG, Schwabach, Germany; 37 °C, 5% $CO_2$ and 95% relative humidity), the MTT (methylthiazolyldiphenyl-tetrazolium bromide) test was done using the method of *Tada et al. (1986)* with modifications (*Matusiewicz et al., 2018*; *Matusiewicz et al., 2019*). Yellow MTT solution is converted to water-insoluble, dark blue MTT formazan, by mitochondrial dehydrogenases of living cells. A total of 15 μL of MTT reagent (Sigma-Aldrich, St. Louis, MO, USA) in PBS (5 mg/mL) was added to the cells and they were incubated (37 °C, 4 h). Then, 100 μL of lysis buffer (10% SDS in 0.01 M HCl) was added and plates were incubated overnight at 37 °C. The absorbance was measured at 570 nm (Infinite M200 microplate reader; Tecan, Männedorf, Switzerland), $n = 6$.
### Effect of fractions of extracts on cell viability (MTT test)

A total of 90 µL of new medium and 10 µL of four fractions: >50 K, 10–50 K, 3–10 K and <3 K of extracts from lyophilized eggs of *H. a. maxima* and *H. a. aspersa* snails, at the concentrations of 1.25 and 0.125 mg/mL were introduced into the cells. The same controls as in Effect of extracts on cell viability (MTT test) section were applied. After 24 h incubation, the MTT test was done, as in Effect of extracts on cell viability (MTT test) section. $n = 4$.

### Effect of extracts on the percentage of live and dead cells (trypan blue test)

The trypan blue test is based on the fact that live cells have intact cell membranes which exclude dyes such as trypan blue, while dead cells take up the dyes (*Strober, 2015*). After mixing the cell suspension with trypan blue, live cells have clear cytoplasms, while dead cells have blue ones. A total of 535 µL of new medium and 59 µL of extracts from lyophilized eggs of *H. a. maxima* and *H. a. aspersa* snails, at the concentration of 25 mg/mL were added to the cells. Equal volumes of sterile deionized water were introduced into the control cells. After 12, 24, 48 and 72 h of incubation in a $CO_2$ incubator (INCO 108 med; Memmert GmbH + Co. KG, Schwabach, Germany; 37 °C, 5% $CO_2$ and 95% relative humidity), the trypan blue test was done. Initially, the cells were trypsinized, washed in cold PBS and centrifuged. The supernatants were removed and the cell pellets were resuspended in PBS (100 µL). Then, 0.4% trypan blue (Sigma-Aldrich, St. Louis, MO, USA; 50 µL) were mixed with the cell suspensions (50 µL). After incubation for 5 min, the samples (10 µL) were loaded into a Bürker chamber. Cells in the entire chamber were photographed using a DMi8 inverted light microscope with a MC190 HD camera, employing the LAS V4.10 software (Leica, Wetzlar, Germany). The results were expressed as the percentage of live and dead cells in the groups treated with the extracts relative to the control group, $n = 5$.

### Effect of extracts on the integrity of cell membranes

Cell membrane damage results in the release into the medium of the cytosolic enzyme, lactate dehydrogenase (LDH). This enzyme can be quantified by applying a coupled enzymatic reaction. It is a catalyst for the transformation of lactate to pyruvate by reduction of $NAD^+$ to NADH. Thereafter, diaphorase utilizes NADH in order to reduce INT tetrazolium salt to red formazan that content is measured (490 nm). The LDH test was carried out as recommended by the commercial kit manufacturer (Thermo Fisher Scientific, Waltham, MA, USA). A total of 100 µL of new medium and 10 µL of extracts from lyophilized eggs of *H. a. maxima* and *H. a. aspersa* snails, at the concentrations of 25, $25 \times 10^{-1}$, $25 \times 10^{-2}$, $25 \times 10^{-3}$, $25 \times 10^{-4}$ and $25 \times 10^{-5}$ mg/mL were added to the cells. Equal volumes of sterile deionized water were added to the control cells. Additional controls were also included. After 24 h and 72 h of incubation in a $CO_2$ incubator (INCO 108 med; Memmert GmbH + Co. KG, Schwabach, Germany; 37 °C, 5% $CO_2$ and 95% relative humidity), LDH activity was evaluated and expressed as % of maximum LDH activity (after cell lysis), $n = 4$.

## Effect of extracts on the content of lipid peroxidation products

A total of 1.5 mL of new medium and 150 µL of extracts from eggs of two snail subspecies, at the concentration of 2.5 mg/mL were introduced into the cells (each of the extracts was added to 18 wells). Equal volumes of sterile deionized water were introduced into the control cells. After 24 h incubation ($CO_2$ incubator INCO 108 med; Memmert GmbH + Co. KG, Schwabach, Germany; 37 °C, 5% $CO_2$ and 95% relative humidity), the cells were trypsinized, washed in cold PBS (two times) and centrifuged. The pellets were resuspended in RIPA buffer (130 µL) and incubated (30 min, 4 °C) to lyse cells. Lysates were centrifuged (14,000× $g$, 10 min, 4 °C), supernatants were collected, frozen in liquid nitrogen and stored at −80 °C. Then, concentration of TBARS in the supernatants was determined as in Determination of lipid peroxidation products section. The total protein concentration in the supernatants was also determined, by the Bradford method, using bovine serum albumin as the standard (*Bradford, 1976*). The results were expressed in µg/mg total protein, $n = 3$.

## Effect of extracts on the types of cell death

A total of 1.5 mL of new medium and 150 µL of extracts from lyophilized eggs of *H. a. maxima* and *H. a. aspersa* snails, at the concentration of 25 mg/mL were added to the cells. Equal volumes of sterile deionized water were added to the control cells. After 24 h of incubation in a $CO_2$ incubator (INCO 108 med; Memmert GmbH + Co. KG, Schwabach, Germany; 37 °C, 5% $CO_2$ and 95% relative humidity), the types of cell death were determined by flow cytometry, in accordance with the procedure of the commercial kit with Alexa Fluor® 488 Annexin V and propidium iodide (PI) (Thermo Fisher Scientific, Waltham, MA, USA). The last one, a fluorescent dye, binds to the nucleic acids staining dead cells and Annexin V conjugated to Alexa Fluor® 488 fluorophore binds to phosphatidyl serine on the external surface of the apoptotic cell membrane. Live cells do not stain, Annexin V-stained cells are considered early apoptotic, PI/Annexin V-stained cells – late apoptotic and PI-stained cells – necrotic. The cells were trypsinized and washed two times in cold PBS. Then, they were centrifuged, the supernatants were removed and the pellets were resuspended in Annexin-binding buffer (100 µL). Alexa Fluor® 488 Annexin V (5 µL) and PI working solution (1 µL) were introduced into the cell suspensions. After incubation of the cells for 15 min (RT), Annexin-binding buffer (400 µL) was introduced, the samples were mixed by pipetting and kept on ice. The cells were assayed using BD FACSCalibur™ flow cytometer (Becton Dickinson, Franklin Lakes, NJ, USA), fluorescence emission intensity was registered by FL1 channel for Alexa Fluor® 488, at 530 nm and FL2 for PI, at 575 nm, with excitation at 488 nm. A total of 10,000 events were registered per sample. Flowing Software 2.5.1 (Perttu Terho, Turku, Finland) was used to generate the plots, $n = 5$.

The cells were photographed using a DMi8 inverted light microscope with a MC190 HD camera, employing the LAS V4.10 software (Leica, Wetzlar, Germany).

## Statistical analysis

The results are shown as the mean ± the SEM (standard error of the mean). The results of antioxidant indicators, content of phenols and total carbohydrates were subjected to a two-way analysis of variance (ANOVA), the mean values were compared by applying the Tukey's post-hoc test. Other results of the chemical composition were submitted to an unpaired Student's t-test. Statgraphics Centurion software (StatPoint Technologies, Inc., Warrenton, VA, USA) was employed. The cell test results were submitted to a one-way ANOVA, the means for groups treated with the extracts were compared to groups treated with deionized water by applying the Dunnett's post-hoc test. Prism 5 software (GraphPad Software Inc., San Diego, CA, USA) was used. The difference between the means at $p < 0.05$ was assumed statistically significant.

# RESULTS

## Antioxidant indicators, content of phenols and total carbohydrates

In the case of eggs of both snail subspecies, the fraction containing particles >50 kDa (K) was marked by the greatest ferric-reducing antioxidant power, several times higher than the value for the extract and fraction with particles 10–50 K (Table 1). The lowest ferric-reducing antioxidant power was shown for fractions containing particles <10 K. The significant subspecies-fraction interaction demonstrated that the fractions comprising particles >50 K were characterized by the greatest ABTS$^{\cdot+}$ scavenging activity. The lower ability to reduce ABTS$^{\cdot+}$ was exhibited by extract from eggs of *H. a. maxima*, followed by extract from eggs of *H. a. aspersa* and fractions containing particles 10–50 K. The lowest ABTS$^{\cdot+}$ scavenging activity was noted in fractions with particles <10 K.

The fractions with particles >50 K had the highest DPPH· scavenging activity. A lower DPPH· scavenging activity was noted for extracts. Fractions containing particles <50 K did not show the ability to reduce the DPPH·.

The statistically significant interaction demonstrated that the content of phenols was the highest in the fractions with particles >50 K. This content was lower in the extracts and the extract from *H. a. maxima* eggs comprised more phenols than the extract from *H. a. aspersa* eggs. Phenol concentration in fractions comprising particles <50 K was slight or no such compounds were detected.

The concentration of total carbohydrates in the lyophilized eggs from *H. a. maxima* was 17.97 ± 0.89%, while in the lyophilized eggs from *H. a. aspersa* – 14.01 ± 0.33%. Significantly the highest concentration of total carbohydrates was observed in the fractions containing particles >50 K. Fractions with particles <50 K contained few carbohydrates.

## Content of crude protein, crude fat, lipid peroxidation products, glutathione, astaxanthin and vitamins

The data showed that lyophilized snail eggs comprised mainly crude protein and its content was similar in the eggs of both subspecies (Table 2). Furthermore, the presence of lipid peroxidation products – TBARS was noted in extracts from eggs. The content of GSH was statistically significantly higher in the eggs of *H. a. maxima* than in *H. a. aspersa*.

**Table 1 Antioxidant indicators and concentration of phenols and total carbohydrates in extracts and fractions of extracts from lyophilized eggs of *Helix aspersa maxima* and *Helix aspersa aspersa*.**

| Factor | Ferric-reducing antioxidant power (mg TROLOX/g) | ABTS·$^{+}$ scavenging activity (mg TROLOX/g) | DPPH· scavenging activity (mg TROLOX/g) | Phenols (mg quercetin/g) | Total carbohydrates (%) |
|---|---|---|---|---|---|
| Subspecies | | | | | |
| *H. a. maxima* | 8.94 ± 3.00 | 0.60 ± 0.17 | 2.03 ± 0.93 | 4.42 ± 2.06 | 12.81 ± 4.85 |
| *H. a. aspersa* | 7.49 ± 2.56 | 0.53 ± 0.16 | 1.63 ± 0.69 | 4.07 ± 2.09 | 11.28 ± 4.42 |
| **Fraction** | | | | | |
| Extract | 7.67 ± 0.55$^{B}$ | 0.92 ± 0.11$^{C}$ | 1.67 ± 0.07$^{A}$ | 4.85 ± 0.46$^{B}$ | 15.99 ± 0.98$^{B}$ |
| >50 K | 27.76 ± 2.88$^{C}$ | 1.61 ± 0.04$^{D}$ | 7.50 ± 1.00$^{B}$ | 16.14 ± 0.14$^{C}$ | 43.77 ± 3.46$^{C}$ |
| 10–50 K | 2.71 ± 0.24$^{AB}$ | 0.25 ± 0.02$^{B}$ | ND | 0.05 ± 0.03$^{A}$ | 0.22 ± 0.03$^{A}$ |
| 3–10 K | 1.42 ± 0.20$^{A}$ | 0.02 ± 0.01$^{A}$ | ND | 0.19 ± 0.12$^{A}$ | 0.20 ± 0.11$^{A}$ |
| <3 K | 1.52 ± 0.10$^{A}$ | 0.03 ± 0.01$^{A}$ | ND | 0.01 ± 0.01$^{A}$ | 0.03 ± 0.01$^{A}$ |
| Subspecies × fraction | | | | | |
| *H. a. maxima* × extract | 8.47 ± 0.64 | 1.10 ± 0.14 | 1.77 ± 0.07 | 5.64 ± 0.20 | 17.97 ± 0.89 |
| *H. a. maxima* × >50 K | 29.94 ± 4.84 | 1.59 ± 0.05 | 8.39 ± 2.00 | 16.04 ± 0.29 | 45.40 ± 6.35 |
| *H. a. maxima* × 10–50 K | 3.14 ± 0.14 | 0.25 ± 0.04 | ND | 0.05 ± 0.05 | 0.24 ± 0.03 |
| *H. a. maxima* × 3–10 K | 1.61 ± 0.28 | 0.03 ± 0.01 | ND | 0.38 ± 0.12 | 0.39 ± 0.16 |
| *H. a. maxima* × <3 K | 1.54 ± 0.12 | 0.02 ± 0.02 | ND | 0.01 ± 0.01 | 0.04 ± 0.02 |
| *H. a. aspersa* × extract | 6.87 ± 0.68 | 0.75 ± 0.10 | 1.57 ± 0.11 | 4.06 ± 0.04 | 14.01 ± 0.33 |
| *H. a. aspersa* × >50 K | 25.58 ± 3.64 | 1.62 ± 0.06 | 6.61 ± 0.45 | 16.24 ± 0.10 | 42.15 ± 4.09 |
| *H. a. aspersa* × 10–50 K | 2.28 ± 0.29 | 0.25 ± 0.02 | ND | 0.05 ± 0.05 | 0.20 ± 0.05 |
| *H. a. aspersa* × 3–10 K | 1.23 ± 0.28 | 0.01 ± 0.01 | ND | ND | 0.01 ± 0.01 |
| *H. a. aspersa* × <3 K | 1.49 ± 0.17 | 0.04 ± 0.00 | ND | ND | 0.02 ± 0.02 |
| Main effects, *p* | | | | | |
| subspecies | 0.254 | 0.123 | 0.344 | 0.001 | 0.328 |
| fraction | <0.001 | <0.001 | <0.001 | <0.001 | <0.001 |
| subspecies × fraction | 0.811 | 0.032 | 0.585 | <0.001 | 0.863 |

**Note:**

Data are expressed as mean ± standard error of the mean. Statistically significant effect: values of one indicator are statistically significantly different when $p < 0.05$, values of one indicator without common superscript (A–D) are statistically significantly different ($p < 0.001$). *n* (number of replicates) = 3 (all indicators except phenols), $n = 2$ (phenols). ABTS·$^{+}$ - 2.2′-azino-bis(3-ethylbenzthiazoline-6-sulfonic acid) radical cation, DPPH· - 2.2-diphenyl-1-picrylhydrazyl radical, K, kDa; ND, not detected.

The presence of crude fat, astaxanthin and vitamins A, C and E was not detected in the eggs of both subspecies. The lyophilized eggs of *H. a. aspersa* contained significantly more vitamin $D_3$.

**Table 2 Content of crude protein, crude fat, thiobarbituric acid reactive substances (TBARS), glutathione (GSH), astaxanthin and vitamins in lyophilized eggs of *Helix aspersa maxima* and *Helix aspersa aspersa*.**

| Parameter | *H. a. maxima* | *H. a. aspersa* | *p* |
|---|---|---|---|
| Crude protein (%) | 29.2 ± 0.0 | 29.0 ± 0.0 | – |
| Crude fat (%) | <0.2 | <0.2 | – |
| TBARS (µg/g) | 0.543 ± 0.026 | 0.481 ± 0.033 | 0.174 |
| GSH (µg/g) | 69.64 ± 1.55 | 44.58 ± 3.15 | <0.001 |
| Astaxanthin (µg/g) | <0.2151 | <0.2151 | – |
| Vitamin A (IU/g) | <0.1566 | <0.1566 | – |
| Vitamin C (µg/g) | <150 | <150 | – |
| Vitamin $D_3$ (µg/g) | 0.4908 ± 0.0060 | 0.8048 ± 0.0028 | <0.001 |
| Vitamin E (µg/g) | <4.5252 | <4.5252 | – |

**Note:**
Data are expressed as mean ± standard error of the mean. Statistically significant effect: values of one indicator are statistically significantly different when $p < 0.05$. $n = 2$ (crude protein, crude fat, vitamin C), $n = 3$ (astaxanthin, vitamins A, $D_3$, E), $n = 6$ (TBARS, GSH).

## Analysis of molecular weights of proteins and glycoproteins

Profiles of proteins and peptides derived from extracts from lyophilized eggs of *H. a. maxima* and *H. a. aspersa* separated by SDS-PAGE are shown in Fig. 1A. The results indicate that the egg proteins and peptides had molecular weights in the range of protein standards: from 10 to 250 kDa. Proteins of molecular weight >55 kDa predominated. The comparison of the protein and peptide profiles shows that *H. a. maxima* eggs (panel (b)) contained more components of low molecular weights compared to *H. a. aspersa* eggs (panel (c)). Protein and peptide bands between 10 and 15 kDa can be observed in the case of both analyzed extracts.

The eggs of *H. a. maxima* (Fig. 1B, panel (c)) and *H. a. aspersa* (Fig. 1B, panel (d)) contained glycoproteins of molecular weights ranging from 8 to 220 kDa. Glycoproteins >50 kDa predominated, especially in the range from 50 to 100 kDa and the profile of glycoproteins was similar in case of both subspecies.

## Analysis of amino acids

The amino acid profiles of lyophilized eggs of *H. a. maxima* and *H. a. aspersa* are presented in Table 3. Eighteen amino acids were identified in eggs, eight of which are essential amino acids (EAA) for humans – leucine (Leu), lysine (Lys), phenylalanine (Phe), valine (Val), threonine (Thr), isoleucine (Ile), methionine (Met) and tryptophan (Trp) (*Wang et al., 2014*). The content of individual EAA, except for Thr and Trp, in the crude protein of the eggs of two snail subspecies differed statistically significantly. AAS, CS and EAAI for snail eggs are shown in Table 4. AAS for amino acids of snail eggs was >1.00, except AAS for Lys and Leu for *H. a. aspersa* eggs. The first limiting amino acid, according to AAS, was His (histidine) for *H. a. maxima* eggs and Lys for *H. a. aspersa* eggs. The contents of Phe + Tyr (tyrosine), Val and Ile in eggs were the highest compared to FAO/WHO standard protein. CS for amino acids of snail eggs was <1.00, except CS for Phe + Tyr and Lys for *H. a. maxima* eggs. The first limiting amino acid, according to CS,

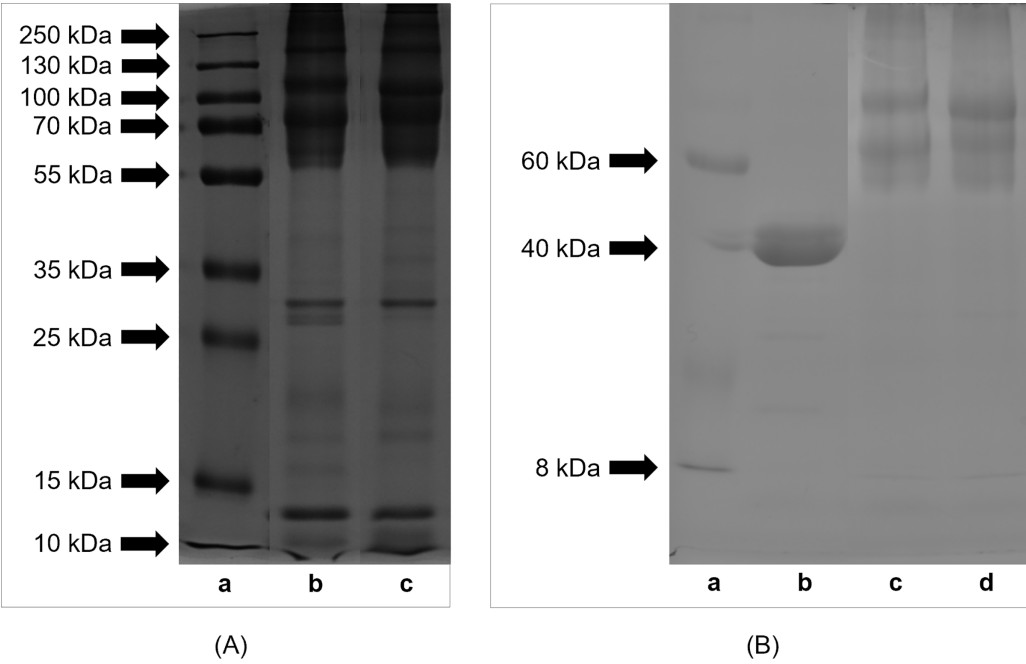

**Figure 1** **SDS-PAGE profile of (A) proteins and (B) glycoproteins isolated from eggs of *Helix aspersa maxima* and *Helix aspersa aspersa*.** (A) Panel (a) – molecular weights of standard proteins (Thermo Fisher Scientific, Waltham, MA, USA); panel (b) – extract from eggs of *H. a. maxima* and panel (c) – extract from eggs of *H. a. aspersa*. (B) panel (a) – molecular weights of standard proteins (Sigma-Aldrich, St. Louis, MO, USA); panel (b) – positive control (horseradish peroxidase); panel (c) – extract from eggs of *H. a. maxima* and panel (d) – extract from eggs of *H. a. aspersa*.

was Met + Cys for eggs of both snail subspecies. The concentration of Met in eggs was low compared to most amino acids (Table 3). The EAAI (FAO/WHO reference amino acid pattern) for snail eggs was >100 and the EAAI (whole egg reference amino acid pattern) was <100 (Table 4). This index had higher value in the case of *H. a. maxima* eggs.

Taking into account non-essential amino acids (NEAA), snail eggs had the highest content of glutamic acid (Glu), aspartic acid (Asp) and serine (Ser) (Table 3).

Snail eggs were characterized by a high total concentration of delicious amino acids (DAA) – Glu, Asp, Ala (alanine) and Gly (glycine). Gly content was the lowest in this group of amino acids.

The concentrations of total amino acids (TAA), EAA, half-essential amino acids (HEAA), NEAA and DAA were higher in the crude protein of *H. a. maxima* eggs than of *H. a. aspersa* eggs. EAA/TAA and EAA/NEAA ratios were higher for *H. a. maxima* eggs and DAA/TAA ratio was higher for *H. a. aspersa* eggs.

## Analysis of minerals

The content of macroelements and microelements, except for Mg, Mn and Se, in the lyophilized eggs of *H. a. maxima* and *H. a. aspersa* snails was statistically significantly different (Table 5). The concentration of macroelements (descending order) in the eggs of both snail subspecies was as follows: Ca, P, Na, K, Mg, S, Cl. The concentration of

**Table 3 Amino acid composition of lyophilized eggs of *Helix aspersa maxima* and *Helix aspersa aspersa* (mg/g crude protein).**

| Amino acids | *H. a. maxima* | *H. a. aspersa* | *p* |
|---|---|---|---|
| Essential amino acids (EAA)* | | | |
| Leucine | 72.84 ± 0.17 | 65.90 ± 0.04 | <0.001 |
| Lysine | 71.73 ± 0.39 | 53.81 ± 0.26 | <0.001 |
| Phenylalanine | 52.33 ± 0.24 | 45.57 ± 0.02 | 0.001 |
| Valine | 51.25 ± 0.36 | 49.48 ± 0.07 | 0.040 |
| Threonine | 45.57 ± 0.06 | 45.90 ± 0.07 | 0.064 |
| Isoleucine | 40.84 ± 0.33 | 38.64 ± 0.09 | 0.023 |
| Methionine | 18.79 ± 0.40 | 15.78 ± 0.10 | 0.018 |
| Tryptophan | 14.49 ± 0.25 | 13.43 ± 0.13 | 0.064 |
| Half-essential amino acids (HEAA)* | | | |
| Arginine | 53.80 ± 0.28 | 44.52 ± 0.45 | 0.003 |
| Histidine | 20.75 ± 0.10 | 20.12 ± 0.23 | 0.123 |
| Non-essential amino acids (NEAA)* | | | |
| Glutamic acid# | 104.54 ± 0.63 | 105.71 ± 1.12 | 0.459 |
| Aspartic acid# | 104.45 ± 0.01 | 90.12 ± 1.19 | 0.007 |
| Serine | 57.00 ± 0.02 | 59.16 ± 0.30 | 0.018 |
| Tyrosine | 49.45 ± 0.48 | 42.98 ± 1.23 | 0.039 |
| Alanine# | 40.62 ± 0.31 | 36.07 ± 0.01 | 0.005 |
| Proline | 38.68 ± 0.05 | 28.21 ± 0.39 | 0.001 |
| Glycine# | 33.26 ± 0.06 | 27.95 ± 0.03 | <0.001 |
| Cysteine | 14.79 ± 0.07 | 14.17 ± 0.42 | 0.281 |
| Amino acid groups and ratios | | | |
| Total amino acids (TAA) | 885.18 ± 0.96 | 797.50 ± 2.95 | 0.001 |
| Essential amino acids (EAA) | 367.84 ± 0.03 | 328.51 ± 0.02 | <0.001 |
| Half-essential amino acids (HEAA) | 74.55 ± 0.37 | 64.63 ± 0.23 | 0.002 |
| Non-essential amino acids (NEAA) | 442.80 ± 0.63 | 404.36 ± 3.19 | 0.007 |
| Delicious amino acids (DAA) | 282.87 ± 1.00 | 259.84 ± 2.29 | 0.012 |
| EAA/TAA | 0.42 | 0.41 | – |
| EAA/NEAA | 0.83 | 0.81 | – |
| DAA/TAA | 0.32 | 0.33 | – |

**Notes:**
Data are expressed as mean ± standard error of the mean.
* For humans.
# Delicious amino acids.
Statistically significant effect: values of one parameter are statistically significantly different when $p < 0.05$. $n = 2$.

microelements (descending order) in *H. a. maxima* eggs was as follows: Cu, Ni, Si, Fe, Mn, Cr, Mo, B, Zn, Co, V, Se, I, Sn and in *H. a. aspersa* eggs: Si, Cu, Fe, Mn, Ni, B, Zn, Mo, Cr, Co, Se, I, V, Sn. No F was detected in the eggs.

## Effect of extracts on cell viability (MTT test)

The influence of various concentrations of water extracts from lyophilized eggs of *H. a. maxima* and *H. a. aspersa* snails on the viability of Caco-2 cells was determined using the

**Table 4 Amino acid score (AAS), chemical score (CS) and essential amino acid index (EAAI) of lyophilized eggs of *Helix aspersa maxima* and *Helix aspersa aspersa*.**

| Amino acids | AAS | | CS | |
|---|---|---|---|---|
| | *H. a. maxima* | *H. a. aspersa* | *H. a. maxima* | *H. a. aspersa* |
| Leucine | 1.10 | 1.00 | 0.85 | 0.77 |
| Lysine | 1.24 | 0.93 | 1.02 | 0.77 |
| Phenylalanine + tyrosine | 1.62 | 1.41 | 1.09 | 0.95 |
| Valine | 1.46 | 1.41 | 0.78 | 0.75 |
| Threonine | 1.34 | 1.35 | 0.97 | 0.98 |
| Isoleucine | 1.46 | 1.38 | 0.76 | 0.72 |
| Methionine + cysteine | 1.34 | 1.20 | 0.59 | 0.53 |
| Tryptophan | 1.32 | 1.22 | 0.85 | 0.79 |
| Histidine | 1.09 | 1.06 | 0.94 | 0.91 |
| EAAI | 132.03 | 120.38 | 85.95 | 78.37 |

**Note:**
Grey fields, the first limiting amino acids.

**Table 5 Elements detected in lyophilized eggs of *Helix aspersa maxima* and *Helix aspersa aspersa*.**

| Elements | *H. a. maxima* | *H. a. aspersa* | *p* |
|---|---|---|---|
| | Macroelements (g/kg) | | |
| Ca | 110 ± 1 | 116 ± 2 | 0.030 |
| P | 5.005 ± 0.070 | 4.776 ± 0.038 | 0.045 |
| Na | 1.835 ± 0.026 | 1.337 ± 0.011 | <0.001 |
| K | 1.714 ± 0.010 | 0.745 ± 0.031 | <0.001 |
| Mg | 0.693 ± 0.004 | 0.685 ± 0.009 | 0.455 |
| S | 0.348 ± 0.002 | 0.374 ± 0.005 | 0.008 |
| Cl | 0.2357 ± 0.0009 | 0.1006 ± 0.0028 | <0.001 |
| | Microelements (mg/kg) | | |
| Cu | 35.5 ± 1.0 | 27.6 ± 0.6 | 0.002 |
| Ni | 25.87 ± 0.62 | 6.15 ± 0.07 | <0.001 |
| Si | 25.0 ± 0.2 | 31.6 ± 0.8 | 0.001 |
| Fe | 23.8 ± 0.7 | 18.5 ± 1.6 | 0.035 |
| Mn | 9.55 ± 0.32 | 10.05 ± 0.05 | 0.201 |
| Cr | 7.75 ± 0.17 | 1.48 ± 0.02 | <0.001 |
| Mo | 7.00 ± 0.05 | 1.65 ± 0.02 | <0.001 |
| B | 5.91 ± 0.05 | 4.95 ± 0.10 | <0.001 |
| Zn | 4.32 ± 0.10 | 2.85 ± 0.04 | <0.001 |
| Co | 0.407 ± 0.013 | 0.255 ± 0.002 | <0.001 |
| V | 0.118 ± 0.002 | 0.040 ± 0.002 | <0.001 |
| Se | 0.109 ± 0.011 | 0.119 ± 0.004 | 0.436 |
| I | 0.026 ± 0.002 | 0.108 ± 0.003 | <0.001 |
| Sn | 0.020 ± 0.000 | 0.023 ± 0.000 | 0.008 |
| F | <10 | <10 | – |

**Note:**
Data are expressed as mean ± standard error of the mean. Statistically significant effect: values of one element are statistically significantly different when $p < 0.05$. $n = 3$.

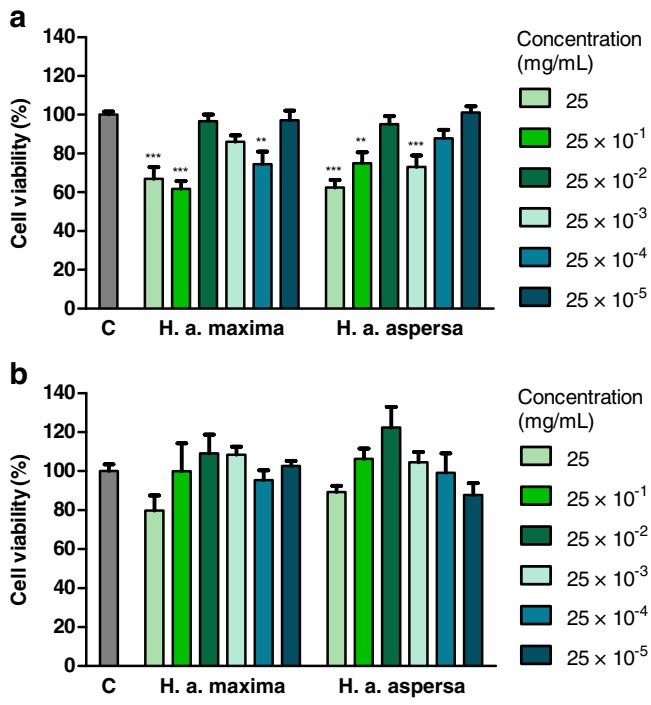

**Figure 2 Viabilty of Caco-2 cells after treatment for (A) 24 h and (B) 72 h with extracts from eggs of *Helix aspersa maxima* and *Helix aspersa aspersa*, at different concentrations.** C indicates control cells (treated with deionized water). Error bars indicate standard error of the mean. Statistically significant effect: two asterisks (**) represent values that differ from control at $p < 0.01$, three asterisks (***) represent values that differ from control at $p < 0.001$. $n = 6$.

MTT test, based on the mitochondrial oxidative activity in live cells. Treatment with an extract from *H. a. maxima* eggs (at concentrations of 25, $25 \times 10^{-1}$ and $25 \times 10^{-4}$ mg/mL) and an extract from *H. a. aspersa* eggs (25, $25 \times 10^{-1}$ and $25 \times 10^{-3}$ mg/mL), for 24 h, resulted in a statistically significant reduction in the viability of Caco-2 colon cancer cells compared to control cells, treated with deionized water (Fig. 2A). Treatment with extracts from eggs of both snail subspecies, at different concentrations, for 72 h, did not statistically significantly affect the viability of Caco-2 cells compared to control cells (Fig. 2B).

### Effect of fractions of extracts on cell viability (MTT test)

Treatment with fraction of an extract from *H. a. maxima* eggs containing particles of a molecular weight <3 kDa (1.25 and 0.125 mg/mL), for 24 h, resulted in a statistically significant reduction of the viability of Caco-2 cells compared to control cells (Fig. 3A). Treatment with fraction of an extract from *H. a. aspersa* eggs containing particles <3 kDa (concentrations as above), for the same time, resulted in a reduction of the viability of Caco-2 cells compared to control cells, but this effect was not statistically significant (Fig. 3B).

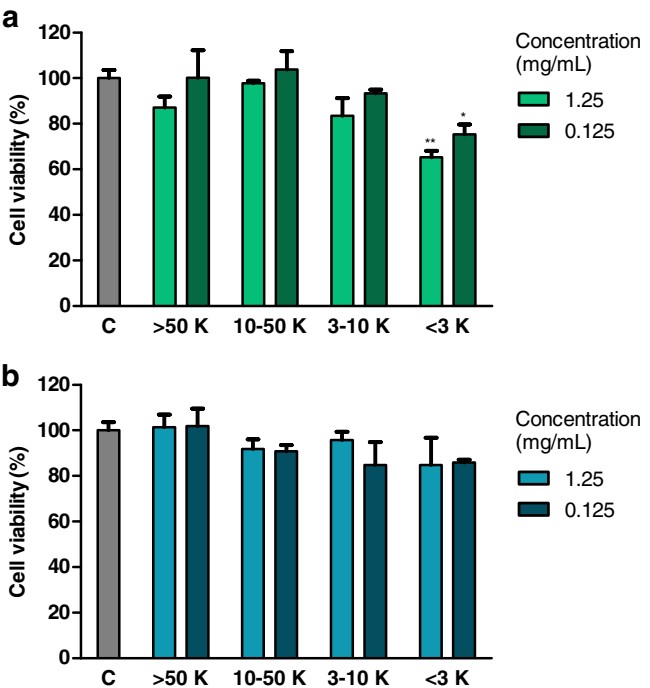

**Figure 3** **Viability of Caco-2 cells after 24 h of treatment with fractions >50 kDa (>50 K), 10–50 kDa (10–50 K), 3–10 kDa (3–10 K) and <3 kDa (<3 K) of extracts from eggs of (A) *Helix aspersa maxima* and (B) *Helix aspersa aspersa*, at two concentrations.** C indicates control cells (treated with deionized water). Error bars indicate standard error of the mean. Statistically significant effect: an asterisk (*) represents values that differ from control at $p < 0.05$, two asterisks (**) represent values that differ from control at $p < 0.01$. $n = 4$.

## Effect of extracts on the percentage of live and dead cells (trypan blue test)

The effect of extracts from eggs of *H. a. maxima* and *H. a. aspersa* on the percentage of live and dead cells was evaluated by the trypan blue assay, in which live cells, having intact cell membranes, exclude trypan blue, while dead cells take up this dye and have blue cytoplasms.

Treatment with extracts from eggs of both snail subspecies (25 mg/mL) for 12 h did not have a statistically significant effect on the percentage of live Caco-2 cells in comparison with control cells, treated with deionized water (Fig. 4A). Incubation with the extract from *H. a. maxima* eggs significantly increased the percentage of dead cells. Treatment with both extracts for 24 h and 48 h resulted in a statistically significant reduction in the percentage of live Caco-2 cells as compared to control cells and did not significantly affect the percentage of dead cells (Figs. 4B and 4C). Treatment with these extracts for 72 h did not significantly affect the percentage of live and dead Caco-2 cells as compared to control cells (Fig. 4D).

## Effect of extracts on the integrity of cell membranes

Treatment with an extract from *H. a. maxima* eggs (25, $25 \times 10^{-1}$ and $25 \times 10^{-3}$ mg/mL) and an extract from *H. a. aspersa* eggs ($25 \times 10^{-1}$ mg/mL), for 24 h, statistically

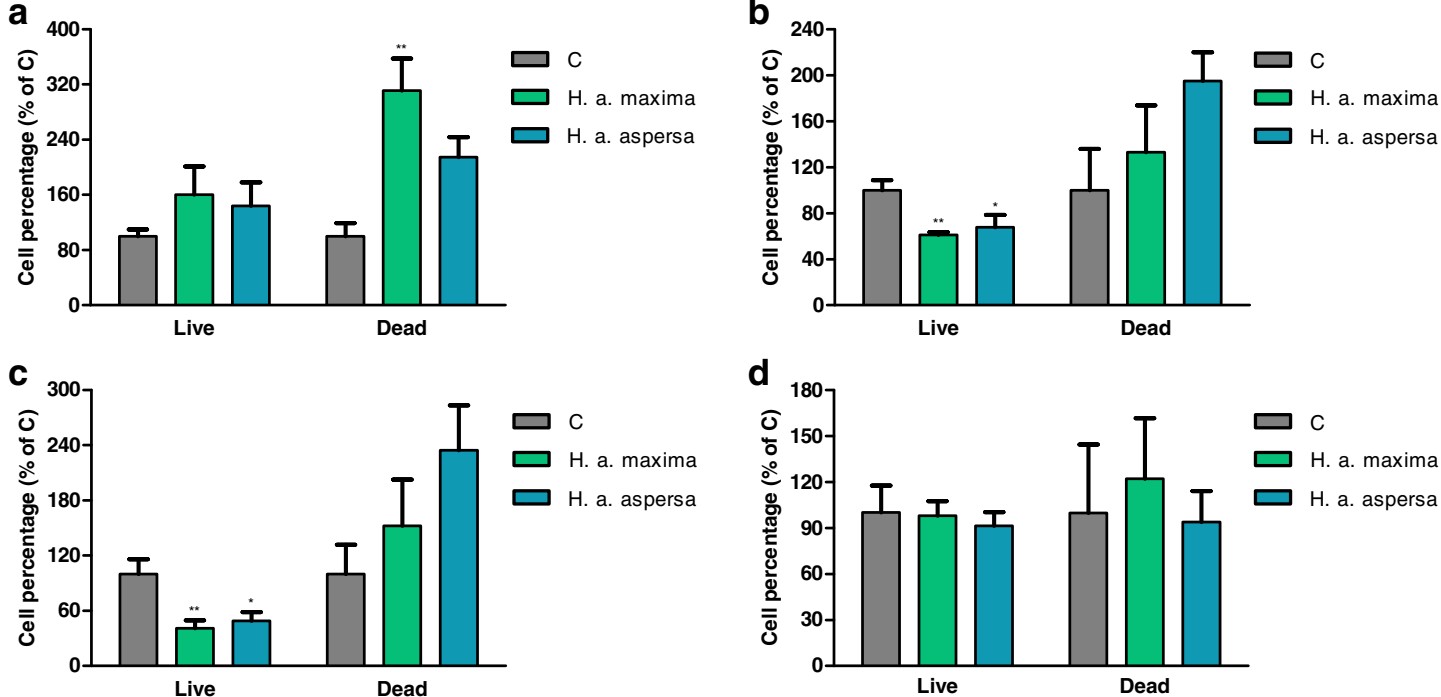

**Figure 4 Percentage of live and dead Caco-2 cells after treatment for (A) 12 h, (B) 24 h, (C) 48 h and (D) 72 h with extracts from eggs of *Helix aspersa maxima* and *Helix aspersa aspersa* (25 mg/mL).** C indicates control cells (treated with deionized water). Error bars indicate standard error of the mean. Statistically significant effect: an asterisk (*) represents values that differ from control at $p < 0.05$, two asterisks (**) represent values that differ from control at $p < 0.01$. $n = 5$.

significantly increased the degree of damage to Caco-2 cell membranes, the activity of the cytosolic enzyme – LDH, released from damaged cells, compared to control cells, treated with deionized water (Fig. 5A). Treatment with extracts from the eggs of both snail subspecies ($25 - 25 \times 10^{-5}$ mg/mL), for 72 h, did not significantly affect the activity of released LDH (Fig. 5B).

## Effect of extracts on the content of lipid peroxidation products

Treatment of cells with extracts from the eggs of both snail subspecies (2.5 mg/mL), for 24 h, increased the amount of lipid peroxidation products – TBARS compared to control cells, treated with deionized water, but this effect turned out to be statistically insignificant (Fig. 6).

## Effect of extracts on the types of cell death

Treatment of Caco-2 cells with extracts from eggs of both snail subspecies (25 mg/mL), for 24 h, caused the induction of apoptosis and reduction of necrosis – an increase in the percentage of early apoptotic cells (effect statistically significant for *H. a. aspersa* egg extract) and a decrease in the percentage of necrotic cells compared to control cells, treated with deionized water (Fig. 7).

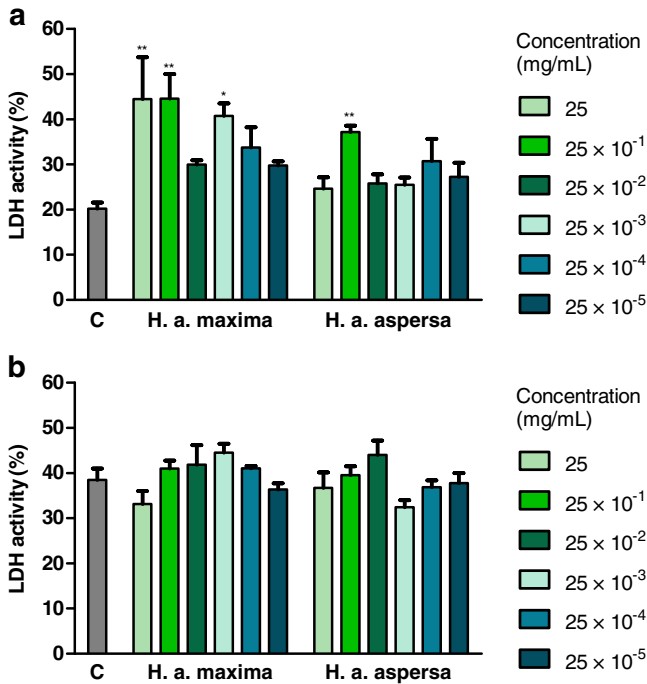

**Figure 5 Integrity of membranes of Caco-2 cells after treatment for (A) 24 h and (B) 72 h with extracts from eggs of *Helix aspersa maxima* and *Helix aspersa aspersa*, at different concentrations.** C indicates control cells (treated with deionized water), LDH – lactate dehydrogenase. Error bars indicate standard error of the mean. Statistically significant effect: an asterisk (*) represents values that differ from control at $p < 0.05$, two asterisks (**) represent values that differ from control at $p < 0.01$. $n = 4$.

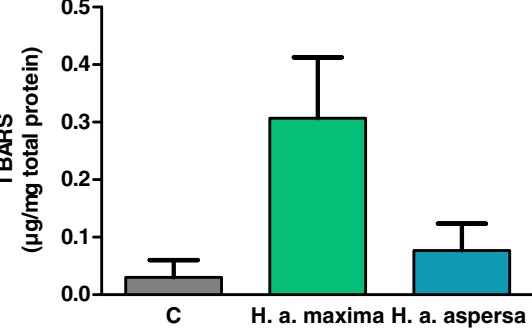

**Figure 6 Concentration of thiobarbituric acid reactive substances (TBARS) produced in Caco-2 cells after treatment for 24 h with extracts from eggs of *Helix aspersa maxima* and *Helix aspersa aspersa*, at the concentration of 2.5 mg/mL.** C indicates control cells (treated with deionized water). Error bars indicate standard error of the mean. $n = 3$.

## DISCUSSION

High concentrations of reactive oxygen species (ROS) in cells may lead to oxidative stress, damage to the cell membranes, proteins and DNA (*Katona & Weiss, 2020*). Antioxidants are necessary in prevention of the formation and inhibition of the activity of ROS (*Dastmalchi et al., 2020*). Some antioxidants and a greater total antioxidant capacity are

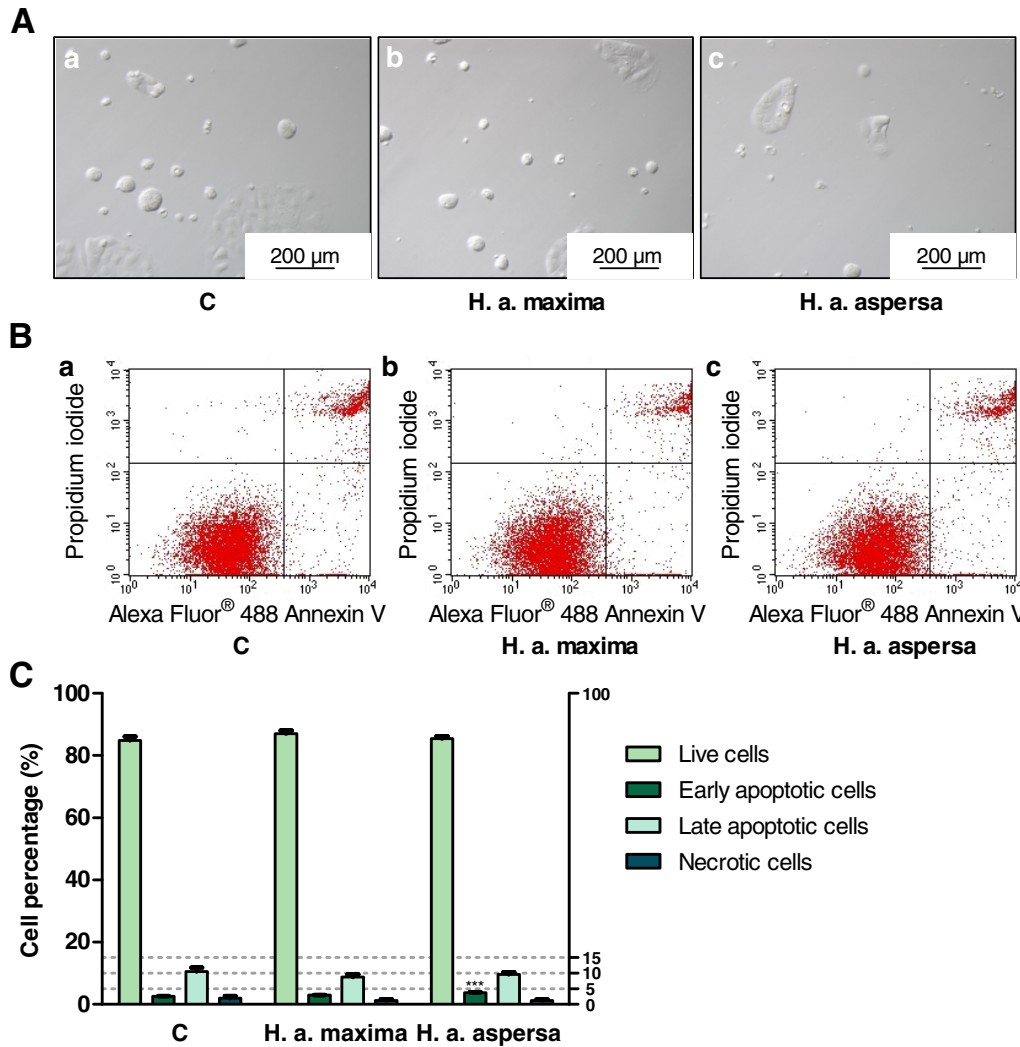

**Figure 7 Types of death of Caco-2 cells after treatment for 24 h with extracts from eggs of *Helix aspersa maxima* and *Helix aspersa aspersa*, at the concentration of 25 mg/mL.** C indicates control cells (treated with deionized water). Error bars indicate standard error of the mean. Statistically significant effect: three asterisks (***) represent values that differ from control at $p < 0.001$. $n = 5$.

related to decreased risk of colorectal cancer (*Katona & Weiss, 2020*; *Chapelle et al., 2020*; *Abbasalizad Farhangi & Vajdi, 2020*). These compounds may be beneficial in the initiation and progression of cancer (*Dastmalchi et al., 2020*). Their action may include a positive effect on cell proliferation, apoptosis, metastasis and drug resistance. In the current research, higher antioxidant activity, expressed as ABTS$^{·+}$ scavenging activity, and phenol content were noted in the extract from lyophilized eggs of *H. a. maxima* than of *H. a. aspersa*. Antioxidants including phenols were concentrated in fractions >50 K.

The differences in the antioxidant potential of the fractions determined with the use of various methods result from the complex kinetics of the contained antioxidants and the time they reach stable endpoints (*Walker & Everette, 2009*). Ferric-reducing antioxidant power and ABTS$^{·+}$ scavenging activity are based on electron donating capacity of bioactive

compounds, and DPPH· scavenging activity is based on electron and hydrogen atom transfer (*Ak & Gülçin, 2008*). ABTS·$^+$ is more reactive than DPPH·. Ferric-reducing antioxidant power differs from the two methods used to determine the concentration of antioxidants, because the reduction of $Fe^{3+}$ to $Fe^{2+}$ is monitored and free radicals are not involved (*Floegel et al., 2011*). The content of hydrophilic and high-pigmented antioxidants is better reflected by ABTS·$^+$ scavenging activity than DPPH· scavenging activity.

Phenols are compounds that can initiate the process of autoxidation under certain conditions, and therefore they can act as prooxidants (*Dai & Mumper, 2010*; *León-González, Auger & Schini-Kerth, 2015*). Such conditions include, for example, high pH, high contents of transition metal ions, $Cu^{2+}$, $Fe^{3+}$ and the presence of oxygen molecules. The high concentration of phenols also favors their prooxidative effect (*León-González, Auger & Schini-Kerth, 2015*). In addition, low molecular weight phenols are easily oxidized and show prooxidative activity, unlike those with high molecular weight, which have little or no prooxidative activity (*Dai & Mumper, 2010*).

Cancer cells are characterized by a higher level of transition metal ions and the mobilization of such endogenous metal ions as Fe and Cu may explain the selective toxicity of polyphenols toward cancer cells (*Hadi et al., 2007*). Moreover, cancer cells of many solid tumors, because of the Warburg effect, are characterized by a high glycolysis level, which leads to a decrease in pH, that exposes Cu bound to chromatin to a potential attack of prooxidants (*Shamim et al., 2012*).

Polyphenol-induced production of ROS may play a major role in apoptosis initiation, even though ROS are also generated as a consequence of it (*Kim et al., 2012*; *Alhosin et al., 2015*; *Liang et al., 2014*; *Khan, Gahlot & Majumdar, 2012*). The prooxidative activity of some polyphenols, in non-cytotoxic concentrations, may sensitize cancer cells to other cancer therapies (*Lee et al., 2014*). Epigallocatechin gallate treatment of chemoresistant HT-29 colon cancer cells was related to an increased ROS production, reduced proliferation and sensitization to 5-fluorouracil (*Hwang et al., 2007*).

ROS induced by phenolic compounds, with/without the presence of transition metals, may contribute to lipid peroxidation (*Oikawa et al., 2003*).

In the study by *Kostadinova et al. (2018)*, the fraction of *H. a. aspersa* mucus containing the smallest particles, <5 kDa, was characterized by a greater antioxidant potential compared to the other fractions tested. Low molecular weight peptides and free amino acids may contribute significantly to the antioxidant potential (*Farvin et al., 2016*). About 70% of the antioxidant peptides had a molecular weight in the range 400–650 Da (*Zou et al., 2016*). In two fractions of *H. a. aspersa* mucus, containing molecules <1 and <3 kDa, similar metabolites with antioxidant activity were detected (*Vassilev et al., 2020*). The antioxidant activity of *H. a. maxima* eggs was demonstrated in the study of *Górka, Oklejewicz & Duda (2017)*.

The concentration of carbohydrates in the lyophilized eggs of the studied snail subspecies was higher than in the lyophilized foot tissues of *H. a. aspersa* snails and similar to their content in the lyophilized mucus of these snails (*Matusiewicz et al., 2018*). Carbohydrates were also concentrated in the fraction >50 K. Other authors' research

shows that the shell of *H. aspersa* eggs consists of crystals of calcium carbonate bound to the mucopolysaccharide matrix that are surrounded by the mucus layer (*Ansart, Madec & Vernon, 2007*; *Tompa, 1976*). Hovingh and Linker showed that chondroitin sulfate is the major mucopolysaccharide (glycosaminoglycan) present in the heart, mantle and kidney of *H. aspersa*, and heparan sulfate was also present in smaller amounts in these organs (*Hovingh & Linker, 1998*). *Wu et al. (2020)* proved that chondroitin sulfate derived from sturgeon (*Acipenser*) reduced the proliferation of HCT 116 human colon carcinoma cells and induced extensive apoptosis. Chondroitin sulfate also inhibited the xenograft HCT 116 tumor development in mice by inhibition of proliferation and induction of apoptosis. Many anticancer drugs based on glycosaminoglycans or their mimetics have been developed with promising results on animal models and in clinical trials (*Morla, 2019*). *Nicolai et al. (2012)* demonstrated that galactogen is the main compound contained in the eggs of *H. a. aspersa* snails, and its concentration turned out to be about six times greater than that of glycogen. In the study of *Górka, Oklejewicz & Duda (2017)* *H. a. maxima* eggs did not contain glucose.

Our research showed that the lyophilized eggs of both snail subspecies consisted mainly of crude protein and were practically fat-free. Protein requirement for maintenance is 0.66 g/kg/d (*Consultation, 2011*). The concentration of crude protein in the lyophilized eggs of the studied snails turned out to be nearly two times lower than in the lyophilized mucus of *H. a. aspersa* and over two and a half times lower than in the lyophilized foot tissues of these snails (*Matusiewicz et al., 2018*).

Murine squamous cell carcinoma VII and human colorectal carcinoma grew slower in mice that were administered low-carbohydrate and high-protein diet comparing to a Western diet – relatively high-carbohydrate and low-protein (*Ho et al., 2011*). This diet reduced glycolysis, on which cancer cells depend to a large extent. In turn, feeding a low-protein diet decreased tumor growth in mouse cancer models (*Rubio-Patiño et al., 2018*).

The lyophilized mucus of *H. a. aspersa* snails comprised almost no crude fat, and the foot tissues contained less than 4% (*Matusiewicz et al., 2018*). The fat comprised in the lyophilized foot tissues, however, had a high nutritional value. The crude protein content in lyophilized eggs of the studied snails was lower than the total protein content in the dry matter of *H. a. maxima* eggs in the study of *Górka, Oklejewicz & Duda (2017)*. In the studies by *Maćkowiak-Dryka, Paszkiewicz & Szkucik (2020)* the fat content in *H. a. maxima* eggs was 0.04%, while in *H. a. aspersa* eggs was 0.03%, however, this fat turned out to have a low nutritional value. Other authors detected the slight presence of triacylglycerols in *H. a. aspersa* eggs (*Nicolai et al., 2012*) and cholesterol in *H. a. aspersa* and *H. a. maxima* eggs (*Górka, Oklejewicz & Duda, 2017*; *Nicolai et al., 2012*).

The concentration of carbohydrates in *H. a. maxima* and *H. a. aspersa* eggs is greater compared to caviar and the raw materials from which other caviar substitutes are obtained (*Maćkowiak-Dryka, Szkucik & Pyz-Łukasik, 2020*; *Maćkowiak-Dryka et al., 2020*). On the other hand, the fat content is much lower and the fat is not a good source of polyunsaturated fatty acids (PUFA).

In our study, in egg extracts were detected lipid peroxidation products.

The product of lipid peroxidation, 4-hydroxynonenal (HNE) can have a stronger cytotoxic, pronecrotic and proapoptotic effect on cancer cells than on normal cells because cancer cells have a lower PUFA content in the cell membrane, to protect against lipid peroxidation (*Andrisic et al., 2018*). It influences oxidative homeostasis and growth regulation. MDA, formed by lipid peroxidation, reacts with bases of nucleic acids and forms adducts that can cause apoptosis (*Cai, Dupertuis & Pichard, 2012*). 4-HNE could suppress growth of Caco-2 and HT-29 cells (*Cai, Dupertuis & Pichard, 2012*). The anticancer action might be related to oxidative stress alterations and consequent apoptosis induction. Glutathione conjugates of HNE are used as chemotherapeutic agents in cancer models.

GSH, a major intracellular antioxidant, regulates cellular redox state, protects cells from lesions induced by lipid peroxides, ROS, reactive nitrogen species, as well as xenobiotics (*Kennedy et al., 2020*). GSH is an important controller of cell apoptosis, ferroptosis, proliferation, differentiation and immune function. Alterations in GSH antioxidant system at the molecular level and disturbances in homeostasis of this compound are related to cancer initiation, progression and response to treatment. Increased GSH concentrations in cancer cells are connected with cancer progression and elevated resistance to chemotherapeutics. In our investigation, *H. a. maxima* eggs contained more GSH than *H. a. aspersa* eggs.

The proteins and peptides of snail eggs had molecular weights from 10 to 250 kDa, and proteins >55 kDa predominated. *H. a. maxima* eggs contained more components of low molecular weights compared to *H. a. aspersa* eggs. Proteins and peptides of a molecular weight between 10 and 15 kDa were present in the eggs of both subspecies.

Sixteen putative cationic and amphipathic anticancer peptides were predicted in two fractions of *A. fulica* mucus, which decreased the viability of MCF7 and Vero cells (*Matusiewicz et al., 2018*; *Teerasak et al., 2016*). Nine of the peptides had a molecular weight <3 kDa, three – in the range of 3–10 kDa, two – in the range of 10–50 kDa and two – >50 kDa. The presence of anticancer peptides was also predicted in the mucus of several mollusks, including land snails (*Tachapuripunya, Roytrakul & Chumnanpuen, 2021*).

The eggs of both snail subspecies presented the similar profile of glycoproteins, of molecular weights from 8 to 220 kDa and glycoproteins >50 kDa predominated, especially in the range from 50 to 100 kDa.

The glycoprotein *Helix aspersa* agglutinin, member of the family of H-type lectins identified in invertebrates, an element of the perivitelline fluid of snail eggs, may be useful in drug delivery systems targeting colorectal cancer, due to binding to the surface of cancer cells (*Pietrzyk-Brzezinska & Bujacz, 2020*).

EAA enable endogenous synthesis of NEAA and their availability is the major limiting factor in protein synthesis (*Bonfili et al., 2017*). AAS for Lys in *H. a. aspersa* lyophilized eggs was <1.00, so this amino acid concentration was lower than in FAO/WHO standard (*FAO/WHO, 1991*). According to this score, the first limiting amino acid of lyophilized *H. a. maxima* eggs was His and of lyophilized *H. a. aspersa* eggs – Lys. For lyophilized foot tissues and mucus of *H. a. aspersa* more than one amino acid had AAS <1.00, with Trp being the first limiting amino acid for foot tissues and Lys for mucus

(*Matusiewicz et al., 2018*). The concentrations of Phe + Tyr, Val and Ile in lyophilized eggs were dominating. In the case of lyophilized foot tissues and mucus of *H. a. aspersa*, these were Ile, Thr and Val (*Matusiewicz et al., 2018*). CS for amino acids of lyophilized snail eggs was <1.00, except CS for Phe + Tyr and Lys for *H. a. maxima* eggs, so their contents were smaller than in whole egg protein pattern. According to this score, the first limiting amino acid of lyophilized eggs was Met + Cys. In the case of lyophilized foot tissues and mucus of *H. a. aspersa*, CS for all amino acids was <1.00, while the first limiting amino acids were for the foot tissues – Met + Cys, and for the mucus – Trp and Met + Cys (*Matusiewicz et al., 2018*). Met content in lyophilized eggs was low comparing to most amino acids. The EAAI (FAO/WHO standard) of lyophilized snail eggs was >100 and the EAAI (whole egg standard) was <100. This index was greater for *H. a. maxima* eggs and significantly greater than for lyophilized foot tissues and mucus of *H. a. aspersa* (*Matusiewicz et al., 2018*).

Dietary restriction of Met may be a main strategy to control cancer growth (*Cavuoto & Fenech, 2012*). Furthermore, Met restriction resulted in killing Met-dependent cancer cells co-cultured with normal cells. Animal research in which diets restricted by Met were studied showed inhibited cancer growth and extend healthy life-span. Met depletion in Met-dependent cancer cells can contribute to cell cycle arrest in late S/G2 phase, the susceptibility of cells to death and their hypersensitivity to chemotherapy.

Glu, Asp and Ser are NEAA found in the highest concentrations in lyophilized snail eggs. Gly content was about two times lower than in the lyophilized foot tissues and mucus of *H. a. aspersa* (*Matusiewicz et al., 2018*). The deprivation of Gly and Ser in the diet inhibits the growth of some cancers, including intestinal (*Sullivan & Vander Heiden, 2017*; *Maddocks et al., 2017*). Depletion of Gly in the diet limits one route for Ser synthesis. Limiting Gly and Ser may reduce the cancer's capability to cope with ROS. In addition, a combination of Gly and Ser deprivation with radiation, a treatment inducing ROS may prove effective. Gly and Ser deprivation limits single-carbon units for nucleotide biosynthesis and this might enhance the effectiveness of drugs that target nucleotide synthesis. Moreover, Gly limited growth of cancer in model animals (*De Mejia & Dia, 2010*).

Branched-chain amino acids (Ile, Leu, Val), administered to obese, diabetic rats, showed the ability to reduce preneoplastic changes.

The ratios of EAA/TAA and EAA/NEAA for lyophilized *H. a. maxima* eggs were higher than for *H. a. aspersa* eggs and these ratios were higher for the examined eggs than for the lyophilized foot tissues and mucus of *H. a. aspersa* (*Matusiewicz et al., 2018*). The effect of EAA and the mixture of amino acids comprised 85% EAA and 15% NEAA on various cancer cells, including Caco-2 and HCT 116, and MCF 10A human breast epithelial cells was examined (*Bonfili et al., 2017*). Both EAA and the mixture containing EAA and NEAA showed antiproliferative and cytotoxic activities, including activation of autophagy and apoptosis. Changing the ratio of EAA to NEAA may be an anticancer strategy leading to the selective cancer cell death. Moreover, EAA administration, with/without the chemotherapeutic drug doxorubicin, increased the mortality of various cancer cells, including HCT 116 (*Corsetti et al., 2015*). EAA increased the concentration of apoptotic

markers. Higher EAA/NEAA ratio may limit the survival of cancer cells and their proliferation.

The lyophilized *H. a. aspersa* eggs contained more vitamin $D_3$ than the lyophilized *H. a. maxima* eggs and both could be a valuable source of this vitamin.

Vitamin $D_3$ is produced mainly in the skin that is exposed to ultraviolet-B radiation of the sun, but can also be acquired from diet and supplements (*Feldman et al., 2014*). In combination with vitamin D binding protein in circulation, vitamin $D_3$ (provitamin cholecalciferol) moves to the liver and is metabolized to 25-hydroxyvitamin D [25-hydroxycholecalciferol, calcidiol, calcifediol, 25(OH)D]. 25(OH)D is transported to the kidneys where it is metabolized to calcitriol [1,25(OH)$_2$D] which is converted to metabolites with less activity.

It seems rational to suggest a vitamin D supplementation at a dose of 800–2,000 IU/day (20–50 μg/day) (*Hanley et al., 2010*).

Vitamin D and calcium affect additively incidence of colorectal adenoma, its malignant transformation and progression (*Huang et al., 2019a*). Calcitriol affects different colon cancer cell lines containing an adequate level of vitamin D receptors (VDR) (*Ferrer-Mayorga et al., 2019*). Its transcription-independent (non-genomic) activities, mediated by extranuclear VDR and alternative receptors, have also been reported. It inhibits proliferation of above cells through different mechanisms and sensitizes them to apoptosis. It induces *CST5*/cystatin D expression which limits proliferation (*Valle et al., 2009*). Calcitriol modulates the expression of histone H3 lysine-27 demethylase Jumonji C domain-containing protein 3 (*JMJD3*) which mediates the activities of calcitriol in colorectal cancer cells including decrease of proliferation (*Pereira et al., 2011*). Calcitriol sensitizes colon cancer cells to apoptosis induction through the increase of expression of proapoptotic genes, the decrease of expression of survival genes and *via* interference with the secretion of IL-1ß by macrophages (*Kaler, Augenlicht & Klampfer, 2009*). VDR agonists enhance the action of chemotherapeutics in animal and cell models of colorectal cancer (*Barbáchano et al., 2018*). Calcitriol inhibits the production of *DKK-4*, promoting chemoresistance (*Pendás-Franco et al., 2008*; *Ebert et al., 2012*). It was demonstrated that *miR*(*microRNA*)-22, induced in a VDR-dependent manner, decreases proliferation of colorectal cancer cells and tumor growth (*Alvarez-Díaz et al., 2012*; *Liu et al., 2018*).

The SUNSHINE trial investigated vitamin $D_3$ supplementation in patients undergoing chemotherapy with unresectable advanced and metastatic colorectal cancer (*Barry, Passarelli & Baron, 2019*; *Ng et al., 2019*). Participants receiving high-dose vitamin $D_3$ (firstly 8,000 IU/day during 2 weeks and then 4,000 IU/day), in comparison to the standard dose (400 IU/day), experienced a 2-month increase (statistically nonsignificant) in median progression-free survival. In adjusted analyses, participants receiving high-dose experienced progression and death less frequently during the 22.9 month-median follow-up. The AMATERASU trial included participants with luminal gastrointestinal cancer, stage I to III, after tumor resection (*Barry, Passarelli & Baron, 2019*; *Urashima et al., 2019*). Patients were supplemented with vitamin $D_3$ (2,000 IU/day) or placebo and a *post hoc* analysis showed a significant benefit from supplementation.

The content of Ca in the lyophilized eggs of *H. a. maxima* and *H. a. aspersa* was about 17 times higher than in the lyophilized foot tissues of *H. a. aspersa* and more than 1.5 times higher than in the lyophilized mucus of this subspecies (*Matusiewicz et al., 2018*). The tested egg lyophilizates comprised about 2 times less P than the lyophilizate from *H. a. aspersa* foot tissues, while they were over 2.5 times richer in this element compared to the lyophilizate from the mucus of these snails (*Matusiewicz et al., 2018*). It was shown that Ca possesses properties against colorectal cancer, it limits proliferation and induces apoptosis (*Zhang & Giovannucci, 2011*). Furthermore, Ca and P demonstrated a protective effect at different adenoma-carcinoma sequence steps (*Kesse et al., 2005*). In turn, *Arnst & Beck (2020)* showed that an excess of inorganic phosphate can influence and promote the cancer phenotype. Higher Ca, K, Mg, Mn, Zn, Se and I intakes and lower P, Na, Cu and Fe intakes may be connected with lower colorectal cancer risk (*Swaminath et al., 2019*; *Meng et al., 2019*). The action of the first group of elements may be related to their antioxidative and other anticarcinogenic effects and the action of the second group of elements – to prooxidative and other procarcinogenic effects. The concentration of Na in lyophilized snail eggs was about 4 times lower than in the lyophilized foot tissues of *H. a. aspersa* and several dozen times lower than in their lyophilized mucus (*Matusiewicz et al., 2018*). The WHO recommends a consumption of Na <2 g/day for adults (*World Health Organization, 2012a*). As for other macroelements, the content of K and Mg in lyophilized eggs turned out to be several – a dozen times lower than in the lyophilized foot tissues and mucus of *H. a. aspersa*, and the content of S and Cl – several – a dozen times lower than in lyophilized foot tissues of these snails (*Matusiewicz et al., 2018*). The WHO recommends the intake of K for adults in an amount of at least 3.51 g/day (*World Health Organization, 2012b*). Recommended nutrient intake (RNI) for Mg for males (19–65 years) is 260 mg/day and for females (19–65 years) – 220 mg/day (*World Health Organization, 2005*). Mg and 25(OH)D may act synergistically in reducing the all-cause mortality risk in colorectal cancer patients (*Fiorentini et al., 2021*). Mg plays the crucial role in biochemical reactions involved in vitamin D synthesis and metabolism. In addition, every 100 mg/day increase in Mg intake was connected with 13% lower colorectal adenoma risk and 12% lower colorectal tumor risk. Moreover, Mg deficiency and high Ca:Mg intake have been linked to a higher colon cancer incidence and mortality. Ca:Mg >2.6–2.8 may have a negative influence on colorectal adenoma outcomes and Ca:Mg <1.7 may have negative consequences, and were connected with an increased total mortality risk (*Uwitonze et al., 2020*). Mg deficiency may increase intracellular Ca concentration promoting ROS generation and it may blunt antioxidant capacity to promote oxidative stress (*Fiorentini et al., 2021*). Cell cultures and animal studies suggest Mg role in carcinogenesis through affecting cell proliferation, apoptosis and other processes (*Zhang et al., 2018*). In turn, *Kuhar et al. (2018)* showed that in colorectal cancer hypomagnesemia is a predictor of the effectiveness of therapy based on anti-epidermal growth factor receptor antibodies. One study showed the capability of S to induce apoptosis of non-small cell lung carcinoma cells that are resistant to drugs (*Saha et al., 2015*). Moreover, S is an element of non-enzymatic antioxidants, cancer therapeutics

(*Mates et al., 2012*). Excess Cl, by reaction with water and mineral compounds, creates trihalomethane, which causes cancer (*Shad et al., 2020*).

The concentration of Cu in the lyophilized *H. a. maxima* eggs was slightly higher compared to the lyophilized *H. a. aspersa* foot tissues, and in lyophilized *H. a. aspersa* eggs – slightly lower (*Matusiewicz et al., 2018*). Other authors showed that Cu was cytotoxic to HT-29 cells, which was connected with activation of apoptosis, increased oxidative stress, alterations in β-oxidation in mitochondria and changes in lipid and energy metabolism (*Xiao et al., 2016*). It was also demonstrated that Cu exerted toxicological effect on Caco-2 cells (*Zödl et al., 2003*). Research on rats indicated that low intake of Cu is a risk factor for the development of colon tumor, induced by 3,2′-dimethyl-4-aminobiphenyl and reduced the activities of Cu,Zn-superoxide dismutase (SOD) and ceruloplasmin (*Davis & Feng, 1999*). According to other authors, progression to colon cancer in rats was related to low Zn concentration and lower Cu,Zn-SOD activity in blood plasma (*Christudoss et al., 2012*). Zn modulates folding and misfolding of p53 which is associated with cancer (*Loh, 2010*). According to others, higher Cu and Zn in blood serum were associated with increased risk of colorectal cancer development. Se, selenoprotein P and Se to Cu ratio were connected with decreased risk (*Cabral et al., 2021*). In turn, Cu metabolism may be a promising target for colorectal cancers harboring KRAS mutations (*Aubert, Nandagopal & Roux, 2020*).

The level of Ni in lyophilized *H. a. aspersa* eggs turned out to be slightly lower compared to lyophilized *H. a. aspersa* foot tissues, and in lyophilized *H. a. maxima* eggs – higher (*Matusiewicz et al., 2018*). Contact with Ni can cause different cancers and oxidative stress and mitochondrial dysfunctions can have a crucial role in its toxicity (*Genchi et al., 2020*). Ni compounds can induce apoptosis in different cancer and normal cells.

The Fe content in lyophilized eggs of both snail subspecies was lower compared to that in lyophilized foot tissues and mucus of *H. a. aspersa* (*Matusiewicz et al., 2018*). RNI for Fe is 19.6–58.8 mg/day for males (18+) and 9.1–27.4 mg/day for females (18+), and is dependent on its bioavailability (*World Health Organization, 2005*).

An excess of intestinal Fe increases the risk of developing colorectal cancer by increasing the proliferation of cancer cells, contributing to colon damage induced by oxidative stress and enhancing oncogenic signaling (*Phipps, Brookes & Al-Hassi, 2021*). Research in rodents showed that increased dietary Fe enhances colonic crypt cell proliferation and colorectal tumor development (*Phipps, Brookes & Al-Hassi, 2021*; *Lund et al., 1998*; *Siegers et al., 1992*). Fe may be able to produce ROS that increase oxidative stress, which causes, among others, lipid peroxidation (*Phipps, Brookes & Al-Hassi, 2021*). Excess oxidative stress can result in colonic inflammation which can lead to colorectal cancer (*Phipps, Brookes & Al-Hassi, 2021*; *Carrier et al., 2001*). In turn, Fe deficiency is widespread in colorectal cancer patients and may result in a diminished immunosurveillance response and changed tumor immune microenvironment, which may potentially lead to cancer progression (*Phipps, Brookes & Al-Hassi, 2021*; *Zohora et al., 2018*). However, *Bird et al. (1996)* showed a U-shaped connection between Fe intake and adenomatous polyps, demonstrating that people who ingest low (<11.6 mg/day) or high (>27.3 mg/day) Fe

amounts had increased colorectal cancer risk, in comparison with people consuming an adequate Fe amount (*Phipps, Brookes & Al-Hassi, 2021*).

Intake of Fe from white meat and plants, containing less heme Fe than red meat, were inversely connected with colorectal cancer risk (*Luo et al., 2019*).

The concentration of Mn in the examined lyophilized snail eggs was higher than in the lyophilized *H. a. aspersa* mucus (*Matusiewicz et al., 2018*). The mitochondrial Mn-SOD inhibited growth of different cancer cells (*Piotrowska, Kucinska & Murias, 2013*). According to other authors, overexpression of Mn-SOD decreased HCT 116 cell growth by inducing cell senescence (*Behrend et al., 2005*). The rats that were administered low Mn in a diet had 23% larger preneoplastic lesions and rats ingesting high Fe in a diet – 18% higher (*Davis & Feng, 1999*).

The concentration of Cr in lyophilized *H. a. maxima* eggs was higher than in lyophilized *H. a. aspersa* foot tissues and mucus, and in lyophilized *H. a. aspersa* eggs – lower (*Matusiewicz et al., 2018*). Cr (III) ameliorated the healing of colitis in mice by promoting antioxidant potential, suppressing ROS and inflammation (*Odukanmi et al., 2017*).

The Mo content in the lyophilized *H. a. maxima* eggs was higher than in the lyophilized foot tissues and mucus of *H. a. aspersa*, and in the lyophilized *H. a. aspersa* eggs – lower (*Matusiewicz et al., 2018*). The incidence and development of induced esophageal tumors were lower in rats which were administered high-Mo diet compared to animals which were fed low-Mo diet (*Komada et al., 1990*). Xanthine oxidase could have played an important function in inhibiting carcinogenesis.

The level of B in lyophilized eggs of both snail subspecies was lower than in lyophilized *H. a. aspersa* foot tissues and mucus (*Matusiewicz et al., 2018*). Low-B diet could increase cancer risk (*Nikkhah & Naghii, 2017*).

The content of Zn in lyophilized eggs was lower compared to lyophilized *H. a. aspersa* foot tissues and mucus (*Matusiewicz et al., 2018*). RNI for Zn for males (19–65 years) is 4.2–14.0 mg/day and for females (19–65 years) is 3.0–9.8 mg/day, and is dependent on its bioavailability (*World Health Organization, 2005*). Zn stabilizes the structure and regulates NF-κB which induces the expression of various genes related to cell proliferation, apoptosis inhibition, resistance to chemotherapeutics and other processes, promotes tumor formation (*Skrajnowska & Bobrowska-Korczak, 2019*). Activation of NF-κB can be blocked by affecting redox status of cells, the application of competitive inhibitors binding to its DNA site, such as Zn or Cr. Zn could protect healthy cells from the cytotoxic and genotoxic effects of $H_2O_2$, but enhances its toxicity in tumor tissue (*Skrajnowska & Bobrowska-Korczak, 2019*; *Sliwinski et al., 2009*). Zn, as an element of CuZn-SOD, has a strong antioxidant potential (*Skrajnowska & Bobrowska-Korczak, 2019*). CuZn-SOD and Mn-SOD affects tumor formation and development (*Skrajnowska & Bobrowska-Korczak, 2019*; *Eapen et al., 1998*; *Liaw et al., 1997*). Another mechanism of the antioxidant activity of Zn comprises its antagonism against minerals participating in lipid peroxidation (Fe and Cu) (*Skrajnowska & Bobrowska-Korczak, 2019*). It also protects the protein -SH groups from oxidation by chelate formation. Zn ions play an important role in the induction of metallothioneins, contributing to cancer development or inhibition. It stabilizes the zinc finger structures, which play important functions in, among others,

proliferation and apoptosis. Zn deficiency may result in a higher cancer initiation and progression risk (*Skrajnowska & Bobrowska-Korczak, 2019*; *Sliwinski et al., 2009*). It plays a major role in the growth and division of cells and programmed death (*Skrajnowska & Bobrowska-Korczak, 2019*). Zn has multidirectional role in the initiation and inhibition of apoptosis. Endogenous Zn is likely indispensable in autophagy induction, under oxidative stress. A meta-analysis of six studies showed that higher Zn intake was associated with reduced colorectal cancer risk (*Li et al., 2014*). Rat studies suggest that Zn deficiency contributes to colorectal cancer development and progression (*Christudoss et al., 2012*). Application of Zn to rats treated with dimethylhydrazine (DMH) reduced tumor incidence, size and multipilicity (*Chadha, Garg & Dhawan, 2010*). Furthermore, increased Zn concentrations inhibited growth of cells that represent various colon cancer stages and induced their death (*John, Briatka & Rudolf, 2011*). The main mechanism turned out to be oxidative stress related to mitochondria disorders and plasma membrane damage. Cell death was cell line dependent; the cells showed signs of apoptosis, necrosis, autophagy and mixed types of cell death.

The concentration of Co in lyophilized eggs of *H. a. maxima* and *H. a. aspersa* was higher in comparison with lyophilized *H. a. aspersa* foot tissues and mucus (*Matusiewicz et al., 2018*). High Co consumption may be associated with an increased colon cancer risk and high Se consumption with decreased risk (*Kiani et al., 2021*).

V was also present in the lyophilized eggs of both snail subspecies. Vanadium N-(2-hydroxy acetophenone) glycinate demonstrated cytotoxic action on human colorectal carcinoma cells and other cancer cells, without toxic influence on normal fibroblasts (*Del Carpio et al., 2018*). It induced apoptosis of human colorectal carcinoma cells by mechanisms including mitochondrial damage and higher ROS level.

The content of Se in lyophilized snail eggs was lower compared to lyophilized *H. a. aspersa* foot tissues and mucus (*Matusiewicz et al., 2018*). RNI for Se for males (19–65 years) is 34 µg/day and for females (19–65 years) is 26 µg/day (*World Health Organization, 2005*). Se is an element of selenocysteine which is part of selenoproteins (SELENO), as glutathione peroxidase (GPX) and thioredoxin reductase (TXNRD) (*Kipp, 2020*). Their important functions include maintaining cellular redox homeostasis. Other selenoproteins, as SELENOF and SELENOS, in turn, participate in the folding and degradation of proteins. There was an inverse correlation between Se intake and colorectal cancer mortality (*Kipp, 2020*; *Schrauzer, White & Schneider, 1977*). Elevated SELENOP level was inversely correlated with colorectal cancer risk (*Kipp, 2020*; *Riboli & Kaaks, 1997*). It seems that suboptimal Se concentration is a cancer risk factor (*Kipp, 2020*; *Clark et al., 1996*). The mechanism of Se's anticancer activity may be increased expression of selenoproteins, which prevent oxidative damage to DNA (*Kipp, 2020*). SELENOH downregulated colorectal cancer cells showed increased growth properties (*Kipp, 2020*; *Bertz et al., 2018*), and loss of GPX2 (*Kipp, 2020*; *Emmink et al., 2014*) and SELENOF (*Kipp, 2020*; *Tsuji et al., 2011*) impaired growth of colorectal cancer cells. The probability of relapse-free survival in colorectal cancer patients was lower in the case of high expression of GPX2 in the tumor (*Kipp, 2020*; *Emmink et al., 2014*). The same relationship was noted for high content of intratumoral SELENOF (*Kipp, 2020*; *Hughes et al., 2018*).

Moreover, the studies demonstrated the anticancer potential of dietary Se by inhibiting the tumor growth of colon cancer cells (*Guardado-Félix et al., 2019*; *Bhattacharya et al., 2011*; *Yoshida et al., 2007*; *Tung et al., 2015*). Its chemopreventive effect could be triggered by various molecular mechanisms including antioxidant protection, cell cycle suppression and apoptosis induction (*Guardado-Félix et al., 2019*; *Fernandes & Gandin, 2015*). A high intake of dietary Se (2.29 μg/g) reduced the growth of tumor of HT-29 colon cancer cells transplanted into immunosuppressed mice and induced cancer cell apoptosis (*Guardado-Félix et al., 2019*). Se was hypothesized to decrease lipid oxidation *via* GPX. In turn, the effect of Se on colorectal cancer in mice, induced by 1,2-dimethylhydrazine, was mediated by its influence on, among others, increasing oxidative stress, MDA concentration and apoptosis (*Ali, Hussein & Kandeil, 2019*). Depending on the concentration, Se can act as an antioxidant and a pro-oxidant in different experimental conditions (*Ali, Hussein & Kandeil, 2019*; *Collery, 2018*; *Uğuz et al., 2009*).

The content of Ca, Mg and Cu in the lyophilized eggs of *H. a. maxima* and *H. a. aspersa* was higher than the content of these elements in the dry matter of *H. a. aspersa* eggs in the Beeby and Richmond study (*Beeby & Richmond, 2001*). The authors also detected the presence of Zn in these eggs. The concentration of Zn and Mn in the lyophilized eggs of *H. a. maxima* and *H. a. aspersa* also turned out to be higher compared to the concentration of these minerals in the dry matter of *H. a. maxima* eggs in the study of *Górka, Oklejewicz & Duda (2017)*, and the content of Ca, Mg, Cu, Fe and Cr in the tested lyophilized snail eggs was similar to the content in the dry matter of *H. a. maxima* eggs.

The extract from *H. a. maxima* eggs (25, $25 \times 10^{-1}$ and $25 \times 10^{-4}$ mg/mL) and the extract from *H. a. aspersa* eggs (25, $25 \times 10^{-1}$ and $25 \times 10^{-3}$ mg/mL) decreased the viability of Caco-2 cells after 24 h of treatment. Cell viability was reduced by fraction of an extract from *H. a. maxima* eggs containing particles of a molecular weight <3 kDa (1.25 and 0.125 mg/mL) and fraction of an extract from *H. a. aspersa* eggs comprising particles of the same molecular weight, at the same concentrations, but the effect was not statistically significant in case of the second extract. The extract from *H. a. maxima* eggs (25 mg/mL), after 12 h of incubation, increased the percentage of dead Caco-2 cells and the extracts from eggs of both snail subspecies, after 24 and 48 h of treatment, reduced the percentage of live cells. In addition, the extract from *H. a. maxima* eggs (25, $25 \times 10^{-1}$ and $25 \times 10^{-3}$ mg/mL) and the extract from *H. a. aspersa* eggs ($25 \times 10^{-1}$ mg/mL), after 24 h of incubation, increased the degree of damage to Caco-2 cell membranes, LDH release. The effective concentrations of extracts depended on the content of individual chemical compounds at the respective concentrations and on their interactions. Lack of statistically significant effect of the same concentrations of extracts on cell viability, percentage of live cells and LDH activity after 72 h compared to the situation after 24 h may have been associated with a different cell density, cell growth rate and secretion of metabolites by the cells (*Schutte et al., 2004*). The composition of the extracts may also have changed during incubation. The extracts from the eggs of both snail subspecies (2.5 mg/mL), after 24 h of incubation, increased the amount of lipid peroxidation products in cells, but this effect turned out to be statistically insignificant. The extracts from eggs (25 mg/mL) caused the induction of apoptosis and reduction of necrosis – an increase in the percentage

of early apoptotic cells (effect statistically insignificant for *H. a. maxima* extract) and a decrease in the percentage of necrotic cells (effect statistically insignificant). The effect of extracts from snail eggs on the cell viability and the induction of apoptosis may be assigned to the presence of antioxidants. The phenols contained in the extracts may have shown a prooxidative activity, influenced on the greater degree of lipid peroxidation in the cells treated with these extracts. The potential cytotoxic activity of phenols, related to the production of ROS may have played an important role in activating apoptosis. Lipid peroxidation products may have also been responsible for the cytotoxic, prooxidative and consequent proapoptotic effects of extracts on Caco-2 cells. Decreased cell viability may be attributed to anticancer peptides, especially those of a molecular weight <3 kDa, which requires further research. The glycoproteins agglutinins, belonging to the H-type lectins family, present in the extracts may have bound to the surface of cancer cells and facilitated the delivery of other compounds to them, which also requires further studies.

The restriction of Met was most likely associated with the reduction of cell viability. Moreover, the appropriate ratio of EAA to NEAA may have also been responsible for limiting the cell viability by the extracts, their cytotoxic and proapoptotic effects. Furthermore, vitamin D may have contributed to the reduction of cell viability and apoptosis. Additionally, Mg may have had a positive effect on the synthesis and metabolism of vitamin D. Ca may have been responsible for limiting viability and inducing apoptosis. S may have contributed to the induction of apoptosis by the extracts.

The cytotoxic effect on cells, related to the activation of apoptosis and the increase of oxidative stress, may have been demonstrated by Cu. Caco-2 cell growth may have been inhibited by Mn, the component of mitochondrial Mn-SOD. Zn may have been responsible for limiting cell viability. It may have contributed to the induction of oxidative stress, causing mitochondria disorders and damage to the cell membranes, to the activation of apoptosis. In turn, Se present in extracts may have increased oxidative stress, the degree of lipid peroxidation and apoptosis. Other bioactive compounds contained in the extracts and their additive and synergistic effects most likely also influenced Caco-2 cells.

## CONCLUSIONS

Extracts from lyophilized eggs of *H. a. maxima* and *H. a. aspersa* snails, due to the content of antioxidants which are necessary in prevention of the formation and inhibition of the activity of ROS, are able to reduce oxidative stress in the body and may be beneficial in the initiation and progression of colorectal cancer. Reduction of the viability of Caco-2 colon cancer cells after application of snail egg extracts, increase of the degree of damage to cell membranes, the amount of lipid peroxidation products generated by cells, induction of apoptosis and reduction of necrosis may be attributed to the presence of antioxidants, phenols, lipid peroxidation products, anticancer peptides, restriction of Met, appropriate ratio of EAA to NEAA, vitamin D, Ca, Mg, S, Cu, Mn, Zn and Se in them. Other bioactive compounds and their additive and synergistic effects most likely also influenced Caco-2 cells. Differences in the effects on cells of the extracts from eggs of two snail subspecies may result from differences in the content of bioactive compounds in them.

It is planned to examine the composition of extracts and fractions containing molecules <3 kDa in terms of the content of peptides with anticancer properties, using mass spectrometry and bioinformatics tools, as well as other elements of the chemical composition of the extracts. It is planned to investigate the effect of extracts and <3 kDa fractions on cell metabolome and to examine other mechanisms of their anticancer activity, and to deepen research on their impact on cell death (apoptosis, necrosis, autophagy).

Natural extracts from snail eggs or the chemical compounds contained in them might be used in the combination therapy of colorectal cancer, targeting many signaling pathways and using a variety of mechanisms to decrease the development of anticancer drug resistance. However, it should be determined whether the extracts/chemical compounds contained in them may sensitize to cytotoxic therapy, intensify the effective concentration of a drug, enhance the combined effects of both therapeutics or exert a cytotoxic effect specifically on cancer cells.

## ACKNOWLEDGEMENTS

This work is part of a habilitation thesis by Magdalena Matusiewicz.

### Funding

This work was supported by the National Science Centre, Poland, "Miniatura" Project, No. 2017/01/X/NZ9/00195. The funders had no role in study design, data collection and analysis, decision to publish, or preparation of the manuscript.

### Grant Disclosures

The following grant information was disclosed by the authors:
National Science Centre, Poland: 2017/01/X/NZ9/00195.

### Competing Interests

The authors declare that they have no competing interests.

### Author Contributions

- Magdalena Matusiewicz conceived and designed the experiments, performed the experiments, analyzed the data, prepared figures and/or tables, authored or reviewed drafts of the paper, and approved the final draft.
- Karolina Marczak performed the experiments, authored or reviewed drafts of the paper, and approved the final draft.
- Barbara Kwiecińska performed the experiments, authored or reviewed drafts of the paper, and approved the final draft.
- Julia Kupis performed the experiments, authored or reviewed drafts of the paper, and approved the final draft.
- Klara Zglińska performed the experiments, authored or reviewed drafts of the paper, and approved the final draft.

- Tomasz Niemiec performed the experiments, authored or reviewed drafts of the paper, and approved the final draft.
- Iwona Kosieradzka conceived and designed the experiments, authored or reviewed drafts of the paper, and approved the final draft.

## Data Availability

The raw data are available in the Supplemental Files.

## Supplemental Information

Supplemental information for this article can be found online at http://dx.doi.org/10.7717/peerj.13217#supplemental-information.

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
