# Peer review of "Effect of extracts from eggs of Helix aspersa maxima and Helix aspersa aspersa snails on Caco-2 colon cancer cells"

_PeerJ, doi:10.7717/peerj.13217_

## Round 0.1 · original submission · Major Revisions

Please address the concerns of the reviewers and amend the manuscript accordingly.

·

Basic reporting

The manuscript meets basic criteria of reporting. Please see comments below for more information.

Experimental design

The manuscript meets all the criteria set out by PeerJ.

Validity of the findings

Please see General Comments section for details.

Additional comments

This paper by Magdalena Matusiewicz et al describes the anticancer effects of extracts (and their fractions) from the eggs of two snail species Helix aspersa maxima and Helix aspersa aspersa on the viability of colorectal adenocarcinoma cell line. The authors have found that the fractions containing <3 kDa from the extracts of Helix aspersa aspersa showed anticancer effects. The extracts reduced the cell viability by inducing apoptotic cell death pathway.
Overall, the manuscript meets the criterion set out by peer J. However, the discussion section of the manuscript is very lengthy and needs to be shortened for publication purposes.

Reviewer 2 ·

Basic reporting

No comment

Experimental design

No comment

Validity of the findings

No comment

Additional comments

1) Please expand MTT test abbreviation in the Abstract (Methods section)

Reviewer 3 ·

Basic reporting

In the paper titled ‘Effect of extracts from eggs of Helix aspersa maxima and Helix aspersa aspersa snails on Caco-2 colon cancer cells’ the authors prepare extracts and fractions from the eggs of two snail sub species and identified components of the extracts. Further they checked the effect of these extracts and fractions on human colorectal cancer cell lines (Caco-2). The authors claim that the extracts have a negative effect on the cancer cell line and could be a potential candidate to be used in combination therapy for cancer treatment. Overall the paper needs to be written properly and the language needs editing . Throughout the paper, in many places, sentences need to be framed properly to help readers understand the content easily. Introduction has to be written more clearly and precisely. The results presented are not conclusive and are not discussed in the paper. In discussion the authors try to just give information related to the paper and fail to discuss the result section. I request the authors to go through the comments and modify the paper before publishing.

Experimental design

See section below

Validity of the findings

1. In the background section, the second line (line 21 and 22), does not correctly reflect what the authors want to communicate. Please rewrite this.
2. Reference missing (Line 59)
3. Reference missing (Line 60)
4. Line 57,58,59- What does transition mean here? Is it an economical/developmental transition? Readers will not be able to understand this sentence if not explained properly.
5. Line 75- What is combination therapy? Is it the use of more than one medication at a time to treat cancer? It is hard to understand what the authors want to explain here.
6. Figure 1A- Even though the authors started with equal weight of eggs for both species, I would recommend a protein quantitation to confirm equal protein concentration before analyzing by SDS PAGE. Also, I recommend the authors to provide a picture of the SDS gel showing bands below 8kDa. This could be easily produced by running the extracts on SDS gel and not allowing the dye front to run away.
7. Figure 1 B needs a known glycoprotein as positive control. What is the sensitivity of this method- Is that glycoproteins which are in lower concentration not stained? Does staining depend on the number of glycan attached to the protein?
8. Line 518 to 524, the result is unexpected with H. a. maxima eggs (at concentrations of 2.5 x 10-2 one does not see reduction in viability but at 2.5 x 10-4 there is significant reduction. Same applies for H. a. aspersa eggs. Also, longer incubation with water extract of eggs do not show any significant reduction in cell viability. This needs an explanation at least in the discussion which is missing.
9. Line 519- the numbers are different in figure (25 x 10-2) and text (2.5 x 10-2). Please correct this.
10. Line 340-Why cells are starved in lower FBS? Does that already make cells sick? What is the logic of using 1% FBS in media?
11. Line 539- treatment with extracts for 72 hours did not affect the percentage of live and dead Caco-2 cells while treatment with 24 affects had a significant effect. What is the explanation for this. Did the authors try 12 hour and 48-hour incubation? What would be the effect if they are treated for 12 and 48 hours?
12. Line 543- Again one does not see an effect with concentration of 25 x 10-2, but 25 x 10-3 gives significant difference. Also, no effect with 72-hour incubation. Any explanation?

Additional comments

1) Discussion is very lengthy starting from page 16 to 27. Most of the information given in the discussion can be removed. Please edit and make it short and focus on discussing the results.

---

## Round 0.2 · accepted · Accept

All critiques were addressed and the revised manuscript is acceptable now.

·

Basic reporting

The manuscript meets all the criteria set out by PeerJ.

Experimental design

The manuscript meets all the criteria set out by PeerJ.

Validity of the findings

The manuscript meets all the criteria set out by PeerJ.

Additional comments

None